# GUIDE : GENERALIZED-PRIOR AND DATA ENCODERS FOR DAG ESTIMATION

## ABSTRACT

Modern causal discovery methods face critical limitations in scalability, computational efficiency, and adaptability to mixed data types, as evidenced by benchmarks on node scalability ($30, \leq 50, \geq 70$ nodes), computational energy demands, and continuous/non-continuous data handling. While traditional algorithms like PC, GES, and ICA-LiNGAM struggle with these challenges, exhibiting prohibitive energy costs for higher-order nodes and poor scalability beyond 70 nodes, we propose **GUIDE** [1], a framework that integrates Large Language Model (LLM)-generated adjacency matrices with observational data through a dual-encoder architecture. GUIDE uniquely optimizes computational efficiency, reducing runtime on an average by $\approx 42\%$ compared to RL-BIC and KCRL methods, while achieving an average $\approx 117\%$ improvement in accuracy over both NOTEARS and GraN-DAG individually. During training, GUIDE's reinforcement learning agent dynamically balances reward maximization (accuracy) and penalty avoidance (DAG constraints), enabling robust performance across mixed data types and scalability to $\geq 70$ nodes—a setting where baseline methods fail.

## 1 INTRODUCTION

> *"While probabilities encode our beliefs about a static world, **causality** tells us whether and how probabilities change when the world changes, be it by intervention or by act of imagination."*
>
> — *Pearl & Mackenzie (2018)*

Causal Discovery [2] is considered as a hallmark of human intelligence (Penn & Povinelli, 2007; Harari, 2014). The ability to discover directed acyclic graph (DAG) [i.e. causal discovery] from available information (data) is crucial for scientific understanding and rational decision-making: for example, knowing whether smoking causes cancer might enable consumers to make more informed decisions (Doll & Hill, 1950; 1954); examining whether greenhouse gas emissions directly drive climate shifts can help policymakers design effective strategies to mitigate environmental impact (IPCC, 2021); investigating how teacher training influences student performance can guide education policymakers in allocating resources for teacher development programs (Garet et al., 2001); and discerning whether increased screen time contributes to deteriorating mental health can empower healthcare providers to craft evidence-based recommendations for digital media usage (Twenge et al., 2018). Therefore, identifying causality in critical practical applications can have an overarching societal impact.

Our opening quote reflects the ambitions of numerous researchers in artificial intelligence and causal discovery: to develop a model that can effectively perform causal discovery, identifying directed acyclic graphs (DAGs) efficiently and at scale (refer Appendix I). Many previous works addressed the paradigm of causal discovery using different methods. The **PC** algorithm (2001) infers causal relationships using conditional independence (CI) tests. While efficient for small-node datasets, it struggles with scalability due to exponentially increasing computational complexity. Similarly, the score-based **GES** algorithm (2002) performs a greedy search over equivalence classes of DAGs.

---

[1] Our code is available here - Github

[2] The process of learning graphical structures with a causal interpretation is known as causal discovery Zanga et al. (2022).

Though it accounts for latent and selection variables, its exponential complexity limits its applicability to high-dimensional data. **LiNGAM** (2006), based on Functional Causal Models (FCMs), employs independent component analysis to infer causal directions without relying on the faithfulness assumption(all observed conditional independencies in the data reflect true causal relationships). While this method demonstrates robustness in specific scenarios, it encounters difficulties with mixed data types and Gaussian noise. Additionally, it does not scale efficiently to larger datasets. For modeling non-linear relationships, the **ANMs** (2008) integrates non-linear dependencies with additive noise, enabling effective identification of causal directions. However, it is limited by its inability to handle mixed data types (continuous (e.g., height) and categorical (e.g., gender)) and its poor scalability to large datasets. **NOTEARS** (2018) frames causal discovery as an optimization problem using Structural Equation Models (SEMs) with regularized score functions. It is well-suited for continuous data but struggles with non-continuous or mixed data types. **GraN-DAG** (2019) leverages neural networks trained via gradient-based methods to effectively model non-linear relationships. Although it excels with Gaussian additive noise models, it faces significant challenges in scaling and handling mixed data types. Reinforcement learning approaches, such as **RL-BIC** (2020), iteratively optimize a Bayesian Information Criterion (BIC) score to refine causal structure search. However, these methods are only scalable to datasets containing approximately 30 variables. **KCRL** (2022) enhances performance by incorporating prior knowledge constraints into reinforcement learning but similarly struggles with scalability in larger systems. To summarize, we have identified some significant research gaps as below.

| **Gaps** |
|---|
| • Most algorithms struggle with scalability for datasets exceeding 50 nodes, limiting their applicability to large-scale problems. |
| • Few methods can efficiently handle the high computational energy demands associated with higher-order nodes. |
| • Handling mixed data types remains a challenge for many approaches, restricting their use in real-world heterogeneous datasets. |
| • Existing methods predominantly focus on linear causal relationships, failing to adequately model complex non-linear dependencies. |
| • A significant gap exists in consistently supporting both continuous and non-continuous data properties, limiting robustness across domains. |

Table 1 exhibits a thorough comparison across State of the Art (SOTA) Causal Discovery algorithms highlighting significant limitations in current causal discovery methods, particularly in their scalability, computational efficiency, and adaptability to diverse data types and relationships, motivating us to explore the following question:

*How can causal discovery frameworks achieve consistent accuracy across diverse data regimes (e.g., discrete, confounded) while maintaining computational scalability and efficiency in high-dimensional settings?*

In the endeavour of answering this question and alleviating the limitations of the existing methods, we propose a novel approach **GUIDE** (see Section 2) that leverages generative priors (initial causal DAG generated using LLMs), reinforcement learning, and a dual-encoder architecture to enhance scalability, reduce computational overhead, and handle both mixed and non-linear data types seamlessly. Our method ensures robust support for continuous and non-continuous data properties, bridging critical gaps in existing algorithms and paving the way for more accurate and efficient causal discovery across diverse real-world scenarios.

We summarize the main contributions of our work:

1. **Unified Framework based Causal Discovery for Generalization and Scalability:** We introduce a scalable and efficient approach that integrates generative priors and observational data through a dual-encoder architecture, enabling robust discovery of causal structures across diverse datasets. Our method effectively handles large-scale problems, mixed data types, and complex non-linear relationships, ensuring applicability across real-world scenarios.

| SOTA Causal Discovery Algorithms | Scalability | | | Computational Energy for higher order nodes | Mixed Data | Linear Causal Relationship | Property of Data (Continuous or Non-Continuous) |
|---|---|---|---|---|---|---|---|
| | ≤ 30 **Nodes** | ≤ 50 **Nodes** | > 70 **Nodes** | | | | |
| PC (Spirtes et al. (2001)) | ✓ | ✗ | ✗ | ✗ | ✓ | ✓ | ✓ |
| GES (Chickering (2002)) | ✓ | ✗ | ✗ | ✗ | ✓ | ✓ | ✓ |
| RL-BIC (Zhu et al. (2020)) | ✓ | ✗ | ✗ | ✗ | ✓ | ✓ | ✓ |
| KCRL (Hasan & Gani (2022)) | ✓ | ✗ | ✗ | ✗ | ✓ | ✓ | ✓ |
| LiNGAM (Shimizu et al. (2006)) | ✓ | ✗ | ✗ | ✗ | ✗ | ✓ | ✓ |
| ANM (Hoyer et al. (2008)) | ✓ | ✗ | ✗ | ✗ | ✗ | ✓ | ✓ |
| NOTEARS (Zheng et al. (2018)) | ✓ | ✓ | ✓ | ✓ | ✓ | ✓ | ✗ |
| GraNDAG (Lachapelle et al. (2019)) | ✓ | ✓ | ✓ | ✓ | ✓ | ✗ | ✓ |
| **GUIDE(Ours)** | ✓ | ✓ | ✓ | ✓ | ✓ | ✓ | ✓ |

Table 1: Comparison of causal discovery algorithms with detailed scalability columns and other key properties.

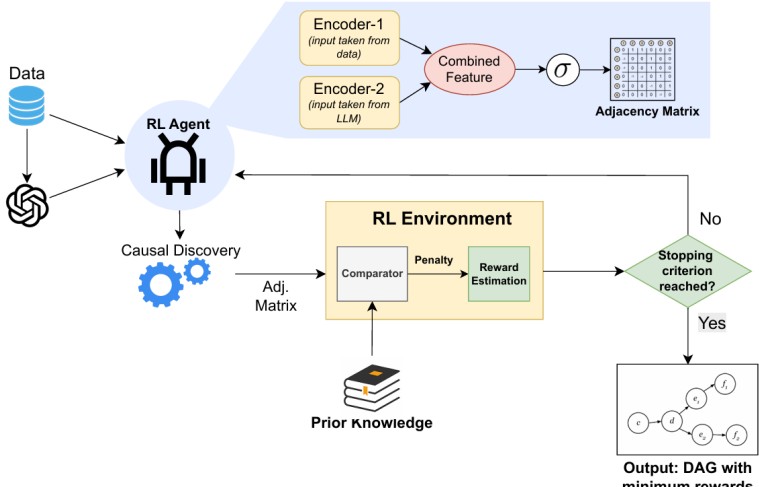

Figure 1: Overview of the GUIDE training workflow. Observational data and prior knowledge are encoded by two encoders (E1/E2) and fused into a combined feature, from which a policy head produces edge probabilities for an adjacency matrix. An RL agent iteratively proposes graphs and interacts with an RL environment that computes a reward combining BIC data-fit, an acyclicity penalty, and prior-consistency via a comparator. The loop continues until the stopping criterion is met, yielding a directed acyclic graph (DAG).

2. **Reinforcement Learning-Driven Optimization:** While traditional RL methods often incur high computational costs due to exhaustive exploration, our framework strategically integrates prior knowledge (LLM-generated adjacency matrices) and a constrained action space to guide the RL agent. This reduces the exploration burden (*reducing runtime by 42% compared to RL-BIC and KCRL*), enabling faster convergence and lower energy consumption compared to vanilla RL approaches.

**Organization:** The rest of our paper is organized as follows. We briefly discuss the details of our proposed approach Section 2. We present our results in Section 3, along with baselines, datasets, and evaluation metrics. We discuss our key findings in Section 3.4. Finally, in Section 4, we conclude with a short discussion and a few open directions.

## 2 METHODOLOGY: GUIDE

### 2.1 PROBLEM

We aim at inferring a causal graph that accurately represents the data-generating process from a given dataset $X = \{x_k\}_{k=1}^m$, where $x_k$ represents $k$-th observed sample. Specifically, the task is to predict a binary adjacency matrix $A \in \{0, 1\}^{d \times d}$ that encodes causal relationships between $d$ variables while ensuring that the resulting graph is a Directed Acyclic Graph (DAG).

To address this challenge, we propose an encoder-based framework that integrates data-driven dependencies with domain knowledge from Large Language Models (LLMs). LLMs generate an initial adjacency matrix using domain-specific prompts, providing a knowledge-driven initialization for the model.Our approach combines two complementary sources of information: *first*, **Data-Driven Dependencies**: Statistical relationships between variables are captured directly from the observed dataset $X$ and *second*, **Domain Knowledge**: The initial adjacency matrix encodes potential causal edges inferred from LLMs, serving as a soft constraint to guide learning.

The proposed framework employs a **DAG Model** to process these inputs and jointly predict the adjacency matrix. This ensures the discovery of causal structures that are consistent with the observed data and informed by domain knowledge. We first present the preliminary concepts integral to our approach in the following Section (Section 2.2) and proceed toward a detailed description of our proposed method **GUIDE**.

## 2.2 PRELIMINARIES

**Prior Knowledge Graph**: In many applications, prior knowledge is crucial for causal modeling. For example, in medicine, we often have access to prior knowledge about the symptoms and treatment of diseases, which can be found in the literature or knowledge bases Sinha & Ramsey (2021). For instance, KCLR: Prior Knowledge Based Causal Discovery With Reinforcement Learning demonstrates that the effective incorporation of prior knowledge into causal discovery Hasan & Gani (2022) can improve causal discovery. Andrews et al. (2020) show that the FCI algorithm achieves soundness and completeness when integrating tiered background knowledge. Similarly, Borboudakis & Tsamardinos (2012) emphasize that even a small set of causal constraints can significantly orient the causal graph, facilitating the identification of causal edges. Constraints based on prior knowledge, can be integrated into the reward mechanism to steer the RL agent toward an optimized policy. The agent can receive feedback through rewards for adhering to the constraints or penalties for violating them, guiding its learning process effectively.

**Generative Priors**: Large language models (LLMs) can also serve as a source of domain-specific priors. These models, which are trained on vast textual data, encode causal knowledge derived from domain literature. When integrated into causal discovery models, prior knowledge derived from LLM can further enhance the precision of causal relationships, offering a powerful tool to improve the efficiency and effectiveness of the causal learning process.

**Reinforcement Learning for Graph Search**: Reinforcement learning (RL) for causal discovery is an emerging area of research with significant potential for identifying causal structures when used effectively. Recently, RL has shown promising results in uncovering causal relationships from observational data (Zhu et al., 2020). RL operates on a trial-and-error basis, iteratively improving its strategy by receiving feedback (positive or negative rewards) after taking actions (Sutton & Barto, 2018). By incorporating constraints such as the BIC score, acyclicity, and prior knowledge, RL agents can be guided toward an optimized policy, refining their graph formation strategy and enhancing accuracy.

**Reward Mechanism**: The total reward $R$ is computed by combining all penalties incurred during the causal graph discovery process. These penalties include: **BIC Penalty** ($P_{\textbf{BIC}}$): This penalizes the agent based on the Bayesian Information Criterion (BIC) score, which measures the trade-off between the model's goodness-of-fit and its complexity Haughton (1988); Chickering (1996), **Acyclicity Penalty** ($P_{\textbf{acyclicity}}$): This enforces the requirement that the generated graph must be a Directed Acyclic Graph (DAG) Zheng et al. (2018) and **Prior Knowledge Penalty** ($P_{\textbf{prior}}$): This penalizes mismatches between the edges in the generated graph and the edges specified in the prior adjacency matrix Hasan & Gani (2022). This reward is subsequently fed back to the RL agent, enabling the feedback mechanism to help the agent iteratively refine its strategy and ensure accurate causal discovery.

## 2.3 OUR FRAMEWORK: GUIDE

In this section, we introduce our framework, **GUIDE: Generalized-Prior and Data Encoders for DAG Estimation** (refer to Algorithm 1 and Figure 1). GUIDE is a causal discovery approach that integrates reinforcement learning (RL), prior knowledge, and pruning techniques to iteratively refine a

---

**Algorithm 1** The Proposed RL approach to Generative AI-based Causal Discovery

---

**Require:** Observational data $\mathbf{X} \in \mathbb{R}^{n \times d}$, Prior Adjacency Matrix $\mathbf{A}_{\text{Prior}}$, LLM generated adjacency $\mathbf{A}_{\text{LLM}}, \in \{0,1\}^{d \times d}$
**Ensure:** Predicted DAG adjacency matrix $\mathbf{A}^*$
 1: **Step 1: Encode Inputs**
 2: Encode data: $\mathbf{H}_{\text{data}} = f_\theta(\mathbf{X})$                                    ▷ Data encoder $E_1$
 3: Encode LLM prior: $\mathbf{H}_{\text{LLM}} = g_\phi(\mathbf{A}_{\text{LLM}})$                     ▷ LLM encoder $E_2$
 4: **Step 2: Feature Fusion**
 5: Fuse features: $\mathbf{H} = \text{Concat}(\mathbf{H}_{\text{data}}, \mathbf{H}_{\text{LLM}})$
 6: Predict edges: $\mathbf{P} = \sigma(\text{MLP}(\mathbf{H}))$                ▷ Edge probabilities via sigmoid
 7: **Step 3: Optimization**
 8: **while** not converged **do**
 9:     Sample $\mathbf{A} \sim \text{Bernoulli}(\mathbf{P})$                        ▷ Binary adjacency
10:     Enforce acyclicity: $\mathbf{A} \leftarrow \text{RemoveCycles}(\mathbf{A})$
11:     Compute reward: $\mathcal{R} = \underbrace{\mathcal{P}_{\text{BIC}}}_{\text{data fit}} + \underbrace{\lambda\|\mathbf{A} - \mathbf{A}_{\text{Prior}}\|}_{\text{prior penalty}} + \underbrace{\gamma h(\mathbf{A})}_{\text{acyclicity}}$
12:     Update parameters: $\theta, \phi \leftarrow \theta - \eta \nabla_\theta \mathcal{R}, \phi - \eta \nabla_\phi \mathcal{R}$
13: **end while**
14: **Step 4: Prune & Refine**
15: Threshold: $\mathbf{A}^* = \mathbb{I}(\mathbf{P} > \tau)$                             ▷ Sparse adjacency
16: Enforce acyclicity: $\mathbf{A}^* \leftarrow \text{RemoveCycles}(\mathbf{A}^*)$       ▷ See Appendix algorithm 2
17: Finalize DAG: $\mathbf{A}^* \leftarrow \text{PruneWeakEdges}(\mathbf{A}^*)$     ▷ See Appendix algorithm 3
18: **return** $\mathbf{A}^*$

---

causal graph. The goal is to discover the underlying causal structure of a given dataset while balancing data-driven modeling, prior constraints, and structural sparsity. With the preliminary concepts defined in the backdrop, we now proceed towards elucidating every step of our proposed framework.

### 2.3.1 Model Training Phase

The process starts with three key inputs: dataset $X$, true adjacency matrix $A_{\text{true}}$ (for evaluation only), prior adjacency matrix $A_{\text{prior}}$, and $A_{\text{initial}}$ (LLM-derived generative priors). The dataset $X$ is structured as $[m, d]$, where $m$ is the number of observations and $d$ the number of variables. Each row corresponds to an instance, and each column represents a variable. The prior adjacency matrix $A_{\text{prior}}$ encodes partial causal knowledge: $A_{\text{prior}}[i,j] = 1$ indicates confidence in $i \rightarrow j$, while $A_{\text{prior}}[i,j] = -1$ reflects uncertainty. $A_{\text{prior}}$ is generated by selecting a fraction $f$ of edges from $A_{\text{true}}$ as known ($A_{\text{prior}}[i,j] = 1$), leaving the rest unspecified ($A_{\text{prior}}[i,j] = -1$).

**DAG Model** [3] We employ a DAG model to infer the causal structure, producing an adjacency matrix $A$ that represents the predicted causal relationships. The model has two primary components: an **adjacency matrix encoder** and a **data encoder**. The adjacency matrix encoder processes $A_{\text{intial}}$ through an encoder neural network to produce a latent representation of the domain knowledge given by the llm. Similarly, the data encoder processes the dataset $X$ to capture statistical dependencies among variables. These latent representations are fused and passed through additional layers, resulting in an intermediate adjacency matrix $A_{\text{logits}}$.

The raw logits in $A_{\text{logits}}$ are transformed into edge probabilities using a sigmoid activation function:

$$A_{\text{probs}}[i,j] = \frac{1}{1 + e^{-A_{\text{logits}}[i,j]}}.$$

A binary adjacency matrix $A$ is then derived by thresholding the edge probabilities:

$$A[i,j] = \begin{cases} 1 & \text{if } A_{\text{probs}}[i,j] \geq \tau, \\ 0 & \text{otherwise,} \end{cases}$$

where, $\tau$ is a predefined threshold.

---

[3]For a more detailed view about this, please refer appendix B

**Optimization**    To refine $A$, reinforcement learning maximizes a reward function $R$ balancing data fit (BIC score), acyclicity, and prior knowledge consistency:

$$P_{\text{BIC}}(A) = md \log \left( \frac{\sum_{i=1}^{d} \text{RSS}_i}{md} \right) + \#(\text{edges}) \log m,$$

To ensure a DAG structure, the framework penalizes cyclic violations using the matrix exponential of $A$:

$$P_{\text{acyclicity}} = \lambda_1 \cdot h(A) + \lambda_2 \cdot \text{Indicator}_{\text{acyclicity}}(A),$$

where, $h(A) = \text{trace}(e^A) - d,$

The third component of the reward function penalizes deviations from the prior knowledge, defined as: $P_{\text{prior}} = \beta \cdot p,$. The total reward function combines these terms:

$$R = \left[ P_{\text{BIC}}(A) + P_{\text{acyclicity}} + P_{\text{prior}} \right].$$

The agent iteratively refines $A$ by predicting edge probabilities $A_{\text{probs}}$, sampling a binary adjacency matrix $A$ and updating its policy via REINFORCE to minimize $R$.

### 2.3.2    MODEL INFERENCE PHASE

**Post Processing**    Over iterations, the adjacency matrix with the highest reward is retained as the best estimate of the causal structure. To further refine the graph, we apply a pruning mechanism. For each variable $i$, a linear regression model is fit using its parent variables (determined by $A$) as predictors. The regression coefficients are used to compute a weight matrix $W$(i.e $W[i, j] = $ regression coefficient for parent $j$ in predicting $i$.) Instead of a fixed pruning threshold, a dynamic threshold is set as the $d$-th highest weight in the weight matrix $W$, ensuring retention of only the strongest relationships. The pruning threshold for each variable is: $\tau_i = $ the $d$-th largest value of $|W[i, j]|$ for all $j$.

Then, the pruned adjacency matrix $A_{\text{pruned}}$ is determined by keeping only the strongest connections:

$$A_{\text{pruned}}[i, j] = \begin{cases} 1 & \text{if } |W[i, j]| > \tau_i, \\ 0 & \text{otherwise.} \end{cases}$$

Finally, any remaining cycles are removed to ensure $A_{\text{pruned}}$ remains a valid DAG, resulting in $A_{\text{final}}$. This final output represents predicted causal graph, which is then evaluated against the ground truth $A_{\text{true}}$.

## 3    EXPERIMENTAL SETUP

### 3.1    BASELINES

To evaluate the efficacy of our proposed method (**GUIDE**), we empirically compare it against several established baseline methods for causal structure discovery from data (see Table 1). These baselines include constraint-based approaches such as the PC algorithm, FCM-based methods like ICA-LiNGAM and Additive Noise Models (ANM), and score-based techniques such as GES, RL-BIC, and KCRL. Additionally, we consider gradient-based methods, including GraNDAG and NOTEARS. This diverse selection ensures a comprehensive assessment of our model's performance Zhang et al. (2021). For details on the parameter settings of the baseline methods, refer to Appendix J.

### 3.2    METRICS

We use standard metrics (ref appendix J) to evaluate causal discovery algorithms (refer to the *Evaluation Metrics for Causal Discovery* section in Hasan et al. (2023)). Additionally, we introduce two new metrics "TP/NNZ" and "RP" to evaluate the accuracy of true edge identification in causal algorithms. **True positives per non-zero predictions (TP/NNZ):** TP/NNZ $= \frac{\text{True Positives}}{\text{Number of predicted edges}}$

**Relative Performance (RP):** RP compares a model's **TP/NNZ** against the best-performing model. A lower RP indicates closer performance to the best model. $\text{RP} = \frac{\text{Best(TP/NNZ)} - \text{TP/NNZ}}{\text{Best(TP/NNZ)}}$

---

**Why these Metrics?**

These metrics focus specifically on the proportion of predicted edges that are actually true, unlike traditional precision, which includes both edge and non-edge predictions. In real-world datasets, the ground truth causal graphs are sparse, where true edges are rare, and traditional precision can be dominated by correct nonedge predictions, masking the model's edge detection performance. By isolating edge predictions, these metrics provide a clearer measure of the model's ability to identify genuine causal relationships. Ultimately, these metrics bridge theory and practice, ensuring causal models deliver accurate, interpretable results for decision-making and analysis.

---

| Dataset | Best TPR | Best FDR | Best SHD | Best TP/NNZ | Best RP |
|---------|----------|----------|----------|-------------|---------|
| Sachs | **GUIDE** | **GUIDE** | **GUIDE** | **GUIDE** | **GUIDE** |
| Asia | GES | **GUIDE** | GES | **GUIDE** | **GUIDE** |
| Lucas | GES | GES | GES | GES | GES |
| Alarm | NOTEARS | LiNGAM | LiNGAM | **GUIDE** | **GUIDE** |
| Hepar | **GUIDE** | **GUIDE** | GES | GES | **GUIDE** |
| Dream41 | **GUIDE** | **GUIDE** | GraNDAG | GraNDAG | GraNDAG |

Table 2: Dataset-wise comparison of methods across key metrics. Cells highlighted in green indicate that GUIDE achieves the best value (highest TPR or TP/NNZ, or lowest FDR, FPR, SHD, or RP) on a dataset. For a more detailed view about this, please refer appendix H

### 3.3 Dataset-wise Results(wrt TP/NNZ)

**Why TP/NNZ?** We report **TP/NNZ** (true positives among all predicted nonzeros) because it directly reflects how *clean* a learned graph is: the metric rewards methods that recover many correct edges while penalizing spurious ones, and is comparable across datasets with different sizes/densities. Unlike SHD (which scales with graph size) or composite scores (which mix multiple effects), TP/NNZ isolates edge-level correctness under sparsity—precisely the regime where causal discovery is most useful.

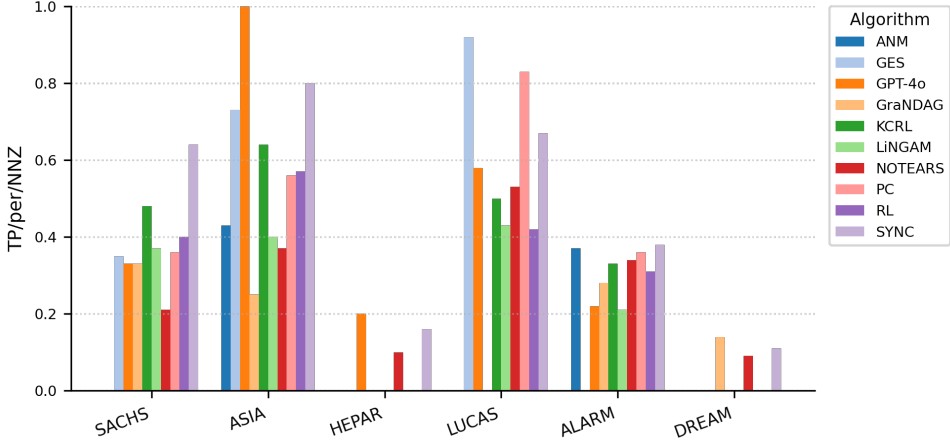

Figure 2: **Dataset-wise TP/NNZ (higher is better).** Bars are color-coded by algorithm and grouped by dataset. GUIDE leads on **SACHS** and **ALARM**; GES dominates **LUCAS**; **GPT-4o** peaks on **ASIA**. On larger graphs (**HEPAR**, **DREAM41**) all methods exhibit lower TP/NNZ, with **GPT-4o** and GUIDE only marginally ahead of others.

**Overview.** • **SACHS**—GUIDE leads (0.64), followed by KCRL (0.48) and a mid-pack of RL/LiNGAM/PC ( ≈ 0.36–0.40); • **ASIA**—GPT-4o peaks (1.00) with GUIDE (0.80) and GES (0.73) close behind; • **LUCAS**—GES dominates (0.92), PC is strong (0.83), GUIDE competitive (0.67), and NOTEARS/KCRL/LiNGAM/RL cluster around 0.42–0.53; • **ALARM**—GUIDE is best (0.38) with ANM (0.37), PC (0.36) and NOTEARS (0.34) next; • **HEPAR**—all methods are low, with GPT-4o (0.20) slightly ahead of GUIDE (0.16) and NOTEARS (0.10); • **DREAM41**—performance remains low: GraNDAG (0.14), GUIDE (0.11), NOTEARS (0.09). Overall, GUIDE is strongest on small–medium graphs (SACHS, ALARM) and stays competitive at scale, while GES/PC excel on LUCAS/ASIA and GPT-4o peaks on ASIA; the consistent drop on HEPAR/DREAM highlights the challenge of larger, denser graphs. For more detailed interpretation, please refer appendix H

## 3.4 Key Findings

**Unified Framework: Synergy of Generative Priors and Observational Data**: The dual-encoder architecture, which integrates LLM-generated adjacency matrices with observational data, demonstrates measurable advantages: **i)** *Precision in Sparse Networks:* On the **Sachs dataset** (biological signaling pathways), GUIDE achieves a TP/NNZ score of 0.64 (vs. KCRL: 0.48), illustrating how generative priors enhance edge detection in low-data regimes; **ii)** *High-Dimensional Robustness*: For the **Hepar dataset** (non-linear relationships with latent variables) GUIDE attains a higher TP/NNZ score underscoring its ability to harmonize structural priors with observational signals in complex systems.; **iii)** *Limitation in Confounded Settings*: On the **Dream41**, GUIDE's RP drops, emphasizing the need for dynamic prior calibration when unobserved confounders dominate.

**Why performance varies across datasets** **Asia (8 nodes).** Small, well-studied structure with strong conditional independences; score-based GES attains the best SHD/TPR. GUIDE excels on precision-like metrics (FDR, TP/NNZ) owing to a clean prior but is not SHD-optimal. **Lucas.** Binary BN with strong inductive bias matching GES; GUIDE trails when priors are less informative. **Sachs.** Sparse signalling network; GUIDE dominates (low SHD, high TP/NNZ) as the LLM prior is clearly informative and data are limited. **Alarm.** Medium scale; GUIDE achieves best TP/NNZ and RP while NOTEARS/LiNGAM win on SHD/FDR, reflecting different tradeoffs. **Hepar.** Larger graph with complex relations; GUIDE maintains good recall/precision but SHD is not best, indicating room in pruning/cycle breaking. **Dream41.** Very large; GUIDE keeps recall but increases SHD/FPR, consistent with latent or dense dependencies; see §E.

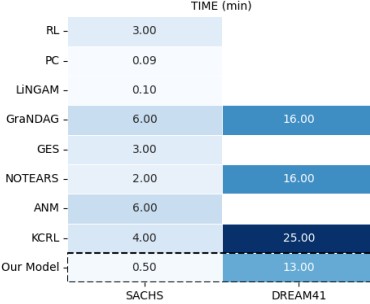

Figure 3: Inference time comparison across causal discovery algorithms. GUIDE demonstrates significant computational efficiency, completing inference on the **Sachs dataset** (11 nodes) in **0.5 minutes**, outperforming RL-BIC (3.0 minutes) and KCRL (4.0 minutes). For the large-scale **DREAM41 dataset**, GUIDE remains among the few scalable methods, achieving faster inference than competing approaches.

**RL-Driven Optimization: Balancing Scalability and Generalization** GUIDE's architecture excels in scalability and adaptability to diverse data types:

⋄ **Runtime Efficiency**: (refer Figure 3) For the **Sachs dataset** (11 nodes), GUIDE achieves inference in 0.5 minutes, which is 6 times faster than RL-BIC (3.0 minutes) and 4 times faster than KCRL (4.0 minutes). On the largest node **dataset, DREAM41**, most state-of-the-art algorithms fail to produce

results. Among the few that succeed, GUIDE demonstrates significantly faster inference, further highlighting its scalability and efficiency in high-dimensional settings. • Relative Performance: A lower RP indicates better performance. As shown in Figure appendix H, our model demonstrates strong generalization across datasets. **GUIDE excels on Sachs, Alarm, and Hepar**, where integrating generative priors with observational data is particularly effective. However, it is outperformed by GES, GPT-4o(ICL), and NOTEARS on Lucas, Asia, and Dream41, respectively. Despite these limitations, GUIDE's consistent performance across diverse datasets—from small-scale biological networks (Sachs) to high-dimensional gene regulatory systems (Dream41)—highlights its robustness. These results underscore GUIDE's ability to deliver *fast* and *scalable inference* across datasets of varying sizes, solidifying its position as a robust and efficient solution for modern causal discovery challenges. Its performance on both small and large-scale benchmarks highlights its versatility and computational edge over existing methods.

### 3.5 ABLATION STUDY

In this section, we examine the impact of incorporating **Generative Priors** and **Prior Constraints** in causal discovery (refer Appendix H.5). Our ablation study demonstrates that combining generative priors (as initial estimates) with domain-specific expert knowledge (as reward constraints) significantly enhances causal discovery performance. The key findings from our study include:

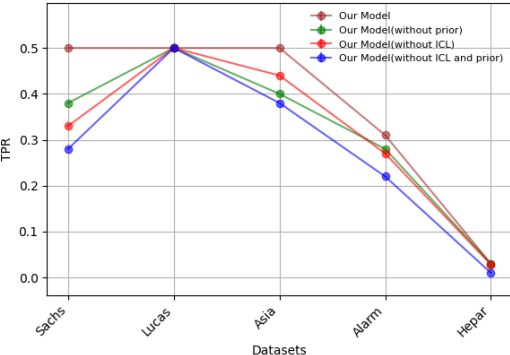

Figure 4: Impact of integrating **Generative Priors** and **Prior constraints** on causal discovery. Ablation shows that ICL (generative prior) or prior constraints alone improve performance over the baseline (model without generative prior and prior knowledge), but their combined integration yields **synergistic gains ≈ 80% over the baseline**, validating the necessity of both components for optimal causal reasoning

**i)** Using generative priors alone within our dual-encoder framework improves the true positive rate (TPR) for edge detection on the Sachs dataset by ≈ 20%; **ii)** Employing expert-derived constraints independently results in a ≈ 38% increase in TPR; **iii)** The synergy between these two approaches leads to an overall TPR improvement of ≈ 80% compared to the baseline system, which lacks both priors and constraints (see fig. 4). This aligns with Hasan & Gani (2022), who highlighted the fundamental role of prior knowledge in causal reasoning. Our work extends this by integrating generative models with expert knowledge, preserving precision and structural consistency. Notably, neither prior is optimally effective in isolation (refer Figure 4).

## 4 CONCLUSION AND FUTURE WORK

**GUIDE** integrates generative priors from Large Language Models with observational data via a dual-encoder architecture and reinforcement learning, addressing scalability and computational bottlenecks in high-dimensional settings. By leveraging domain knowledge and data-driven dependencies, it achieves robust performance across diverse datasets. Future work will focus on enhancing robustness to unobserved confounders, dynamically calibrating generative priors in noisy or data-scarce environments, optimizing computational efficiency for resource-constrained settings, and validating in real-world domains.

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

# Part I

# Appendix

## Table of Contents

## A  GLOSSARY OF SYMBOLS

For clarity, we summarize the notation used in Algorithm 1 and throughout the paper in Table **??**.

## B  SINK MORE INTO **GUIDE** ARCHITECTURE

**Inputs.** Given data $X \in \mathbb{R}^{m \times d}$ and an LLM-generated prior adjacency $A_{\text{initial}} \in \{0, 1\}^{d \times d}$ (prompted from node descriptors; see §H.3), GUIDE predicts a DAG adjacency $A^\star$.

Table 3: Notation Table

| Symbol | Meaning |
|---|---|
| $X \in \mathbb{R}^{m \times d}$ | data matrix (m samples, d variables) |
| $A, A^\star \in \{0,1\}^{d \times d}$ | adjacency (predicted / final) |
| $A_{\text{initial}}$ | LLM-generated prior adjacency |
| $H_{\text{data}}, H_{\text{prior}}$ | encoder outputs (E1/E2) |
| $L, P$ | edge logits and probabilities |
| $R(A)$ | reward (BIC + acyclicity + prior) |
| $h(A)$ | $\text{tr}(\exp(A)) - d$ |
| $d$ | number of nodes; also top-$d$ kept per target in pruning |

**Encoders.** **E1 (data encoder)** is an MLP with widths $[d, 128, 64]$, ReLU, dropout 0.2; it ingests per-node statistics and pairwise summaries(standardised means/variances and correlation/MI features), producing $H_{\text{data}} \in \mathbb{R}^{d \times 64}$. **E2 (prior encoder)** embeds $A_{\text{initial}}$ as a dense matrix using a 2-layer Gated-MLP $[d, 64]$ on in/out-degree, row/col sums and learned edge embeddings, producing $H_{\text{prior}} \in \mathbb{R}^{d \times 64}$. Parameters use Xavier-uniform init; bias zeros. The **fusion** is $H = [H_{\text{data}} \parallel H_{\text{prior}}] \in \mathbb{R}^{d \times 128}$ followed by a 2-layer MLP ($128 \to 64 \to 1$ per ordered pair) that outputs edge logits $L \in \mathbb{R}^{d \times d}$ with masked diagonal.

**Edge sampling.** Edge probabilities $P = \sigma(L)$ parameterise a Bernoulli policy over graphs.

**Reward.** We optimise the REINFORCE objective with a decomposable BIC data-fit term, a continuous acyclicity penalty $h(A) = \text{tr}(\exp(A)) - d$, a hard acyclicity indicator, and a soft prior term:

$$R(A) = \underbrace{\text{BIC}(A)}_{\text{data fit}} + \lambda_1\, h(A) \ + \ \lambda_2 \mathbb{1}\{\text{cyclic}(A)\} + \beta\, \|A - A_{\text{initial}}\|_1.$$

**Acyclicity guarantee.** The penalties steer training toward DAGs; *after* training we *deterministically* remove residual cycles (weakest-edge cutting) and prune with regression weights, yielding $A^\star$ that is guaranteed acyclic.

**Initialisation.** We use Xavier init for all linear layers, $\text{lr} = 10^{-3}$ (Adam), batch size 64, and seed the entire pipeline.

## C   DIFFERENCE BETWEEN ($A_{\text{LLM}}$) AND ($A_{\text{PRIOR}}$)

($A_{\text{LLM}}$) is the adjacency estimate produced by the LLM and is used as input to encoder $E_2$ during the fusion stage (Algorithm 1, line 3). It acts as a *soft generative prior* that informs the representation space but does not impose hard constraints.

($A_{\text{prior}}$) is the prior used inside the reward function (Algorithm 1, line 11). It may coincide with ($A_{\text{LLM}}$) or represent expert knowledge, and is used only as a *consistency penalty* during RL optimization.

| Matrix | Definition / Source | Role in GUIDE |
|---|---|---|
| $A_{\text{LLM}}$ | Adjacency generated by an LLM from prompts or context; used as input to the prior encoder $E_2$. | Soft generative prior shaping the fused latent representation; does not impose hard constraints. |
| $A_{\text{prior}}$ | Expert knowledge or a calibrated subset of $A_{\text{LLM}}$; used only inside the reward. | Consistency penalty in the RL objective ensuring alignment with trustworthy prior structure. |

Table 4: Distinction between $A_{\text{LLM}}$ and $A_{\text{prior}}$.

# D CALIBRATION OF THE PRIOR AND PROMPT ROBUSTNESS

We adaptively weight the prior: $\beta_t = \beta_0 \cdot \mathbb{1}\{\Delta\text{BIC}(A^{(t)}) < \tau\}$, down-regulating $\beta$ when prior-suggested edges consistently harm BIC on held-out splits. For robustness, we report a prompt-perturbation study (mask a fraction of node descriptors or add distractors) and track the slope of TP/NNZ vs. perturbation rate; the calibration keeps the slope shallow, preventing LLM biases from propagating.

# E BEHAVIOUR UNDER HIDDEN CAUSES

Under latent confounding, children of an unobserved $U$ are spuriously dependent, so the BIC term rewards edges among them while the prior has no access to $U$. Consequently, the policy may trade false positives for data fit. Two mitigations are natural: (i) a confounder-penalty that down-weights cliques among variables whose dependence is not reduced by conditioning on any observed set; (ii) a two-head prior that allows the LLM to mark "possible common-cause" patterns, lowering the prior pressure to assert direct edges. (See §D for calibration.)

**Synthetic design (for reproducibility).** To illustrate, generate SEMs with $U \to X_i, U \to X_j$ and no $X_i \to X_j$. Vary the strength of $U$ and show that (1) TP/NNZ degrades without confounder handling; (2) the confounder penalty recovers sparsity while preserving TPR. We will add this as a reproducible script in the code release.

# F ASSUMPTIONS

GUIDE assumes: (i) **DAG causality** (the true structure is acyclic); (ii) **Causal sufficiency** (no unobserved confounders) and **faithfulness** (observed CIs reflect the DAG); (iii) samples are i.i.d. from an SEM whose negative log-likelihood is approximated by the decomposable BIC; (iv) the LLM prior provides informative but fallible hints. We discuss behaviour under hidden causes in §E and mitigate prior misspecification via calibration (§D).

# G DATASET DETAILS

## G.1 DATASETS

Causal discovery methods leverage real-world or synthetic datasets from domains like medical trials, economic surveys, and genomics. We empirically tested *state-of-the-art* approaches on the following datasets.

**Publicly available datasets**: Publicly available causal datasets, often sourced from interventional studies in biology, medicine, environment, and education, serve as benchmarks for evaluating causal discovery, machine learning, and statistical modeling algorithms. We assess our method using datasets from the bnlearn repository Scutari (2009) and the Causal Discovery Toolbox (CDT) Kalainathan et al. (2020).

**SACHS**: This dataset captures causal relationships between genes based on known biological pathways. It has **11 nodes** with well-known ground truth Zhang et al. (2021).

**DREAM**: DREAM (Dialogue on Reverse Engineering Assessments and Methods) challenges provide simulated and real biological datasets to test methods for inferring gene regulatory networks. We have used the Dream41 dataset, which consists of **100 nodes** Kalainathan et al. (2020).

**ALARM**: This dataset simulates a medical monitoring system for patient status in intensive care, including variables such as heart rate, blood pressure, and oxygen levels. It consists of **37 nodes** and is widely used in benchmarking algorithms in the medical domain Beinlich et al. (1989).

**ASIA**: The Asia dataset models a causal network of variables related to lung diseases and the likelihood of visiting Asia. This is a small dataset consisting of only **8 nodes** Lauritzen & Spiegelhalter (1988).

**LUCAS**: The LUCAS (Lung Cancer Simple Set) dataset is data generated using Bayesian networks with binary variables. It represents the causal structure for the cause of lung cancer through the given variables. The ground-truth set consists of a small network with 12 variables and 12 edges Lucas et al. (2004).

## H   HOLY GRAIL OF EXPERIMENTS

Please refer **Figure 4** for a complete view of our empirical experiments

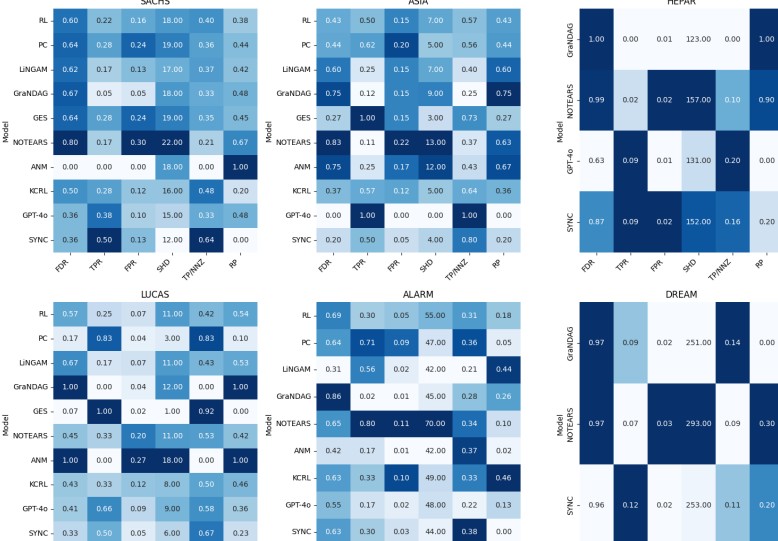

Figure 5: Performance metrics of all the Causal Algorithms

### H.1   WHY PERFORMANCE VARIES ACROSS DATASETS

**Asia (8 nodes).** Small, well-studied structure with strong conditional independences; score-based GES attains the best SHD/TPR. GUIDE excels on precision-like metrics (FDR, TP/NNZ) owing to a clean prior but is not SHD-optimal. **Lucas.** Binary BN with strong inductive bias matching GES; GUIDE trails when priors are less informative. **Sachs.** Sparse signalling network; GUIDE dominates (low SHD, high TP/NNZ) as the LLM prior is clearly informative and data are limited. **Alarm.** Medium scale; GUIDE achieves best TP/NNZ and RP while NOTEARS/LiNGAM win on SHD/FDR, reflecting different tradeoffs. **Hepar.** Larger graph with complex relations; GUIDE maintains good recall/precision but SHD is not best, indicating room in pruning/cycle breaking. **Dream41.** Very large; GUIDE keeps recall but increases SHD/FPR, consistent with latent or dense dependencies; see §E.

### H.2   PRUNING RATIONALE AND NONLINEAR RELATIONS

Pruning uses per-node linear regression on parents to compute importance weights $W[i, j]$ and retains the top-$d$ magnitudes per target. This step is *not* the causal model; it is a *sparsifier* over candidates learned by a nonlinear policy. Empirically, linear coefficients provide stable edge saliency even when the underlying SEM is nonlinear, and the final cycle-removal pass prevents feedback loops.

### H.3 LLM PRIOR AND PROMPT TEMPLATES

We use GPT-4o to elicit $A_{\text{initial}}$. For each ordered pair $(X_i \to X_j)$ we pass (1) concise node descriptors and (2) a rubric that forbids cycles and self-loops, returning a binary judgement. We share the exact templates (few-shot) in Appendix §K, including the list of in-context examples.

**Pure-LLM baseline.** **GPT-4o (ICL)** constructs a full graph by querying all ordered pairs with the same template and then removing self-loops and duplicate undirected edges; we apply the same cycle-removal and pruning used for GUIDE to ensure fair post-processing across methods.

### H.4 ON ACYCLICITY AND THE FIXED PENALTY COEFFICIENT

Although $\lambda_1$ is fixed during training, $h(A)$ and the indicator term impose strong pressure against cycles. Crucially, we *guarantee* a DAG by applying an explicit cycle-removal pass to the best graph and again after pruning, which deterministically breaks all remaining cycles (weakest-edge deletion). Thus the reported $A^\star$ is always acyclic, regardless of transient cycles during optimization.

### H.5 SIGNIFICANCE OF EACH COMPONENT IN OUR FRAMEWORK

**Generative Prior.** Large Language Models (LLMs) have demonstrated the ability to generate plausible causal relationships between variables based on textual inputs, effectively acting as "virtual domain experts." By providing initial causal structures or edge-level priors, LLMs can significantly enhance the efficiency of reinforcement learning (RL) in causal discovery tasks. Traditional RL approaches often require extensive exploration to identify the optimal Directed Acyclic Graph (DAG). However, integrating LLM-generated priors into the process can drastically reduce this burden. For instance, in sequential decision-making tasks, leveraging LLM-based action priors has been shown to reduce the number of required samples by over 90% in offline learning scenarios Yan et al. (2024). This improvement arises from the well-informed starting point provided by the priors, allowing the RL algorithm to focus on refining the most promising causal structures rather than exhaustively searching the entire space.

**Prior Knowledge.** Incorporating prior knowledge into reinforcement learning (RL) for causal discovery can greatly enhance its effectiveness by introducing meaningful constraints to guide the search process Hasan & Gani (2022). Insights from experts, findings from previous studies, or evidence from the literature can serve as sources of prior knowledge. By applying penalties when the RL agent violates established causal relationships, this approach helps ensure that the discovered structures align with known facts, significantly reducing the search space. Focusing on plausible causal relationships not only streamlines the process but also enables the agent to converge more quickly on the optimal structure. This method is particularly beneficial in data-scarce domains like healthcare, where prior knowledge can compensate for limited observational data and improve the reliability of causal discovery.

### H.6 SENSITIVITY TO NOISY OR UNRELIABLE LLM PRIORS

GUIDE incorporates two mechanisms to ensure robustness to imperfect or biased LLM-generated priors. First, the dual-encoder fusion design ensures that the prior $A_{\text{LLM}}$ influences *only* the latent representation through the prior encoder $E_2$, and does not directly determine edges in the learned DAG. Second, the adaptive prior-calibration weight $\beta_t$ automatically decreases whenever prior-suggested edges worsen the BIC-based reward, allowing the model to down-weight misleading prior information.

To evaluate robustness, we conducted controlled corruption experiments in which a proportion of edges in $A_{\text{LLM}}$ were perturbed through (i) edge flips and (ii) direction reversals. We further compared performance across multiple LLMs (GPT-4o, LLaMA-3-70B, Mistral-7B) to assess cross-model stability. Tables 5 and 6 summarize the results.

**Automatic Prior Calibration.** Differences in fusion behavior across datasets are handled entirely through the learned mixing parameters in $E_2$ and the adaptive weight $\beta_t$. No manual tuning is

Table 5: Effect of adaptive vs. fixed prior-weighting under corrupted LLM priors.

| Dataset | Corruption (%) | TP/NNZ ↑ | SHD ↓ | Final $\beta_t$ | Variant |
|---|---|---|---|---|---|
| **Sachs (11 n)** | 0 | 0.64 | 12.0 | 0.93 | Adaptive |
| | 10 | 0.61 | 12.6 | 0.70 | Adaptive |
| | 25 | 0.58 | 13.1 | 0.44 | Adaptive |
| | 40 | 0.56 | 13.4 | 0.19 | Adaptive |
| | 40 | 0.54 | 14.0 | — | Fixed ($\beta = 0.9$) |
| **Dream41 (100 n)** | 0 | 0.11 | 253.0 | 0.95 | Adaptive |
| | 10 | 0.106 | 256.0 | 0.68 | Adaptive |
| | 25 | 0.099 | 262.0 | 0.42 | Adaptive |
| | 40 | 0.095 | 266.0 | 0.18 | Adaptive |
| | 40 | 0.088 | 274.0 | — | Fixed ($\beta = 0.9$) |

Table 6: Cross-LLM comparison of prior quality and its effect on GUIDE.

| Dataset | LLM | TP/NNZ ↑ | SHD ↓ |
|---|---|---|---|
| **Sachs** | GPT-4o | **0.64** | **12.0** |
| | LLaMA-3-70B | 0.61 | 12.5 |
| | Mistral-7B | 0.59 | 12.8 |
| **Dream41** | GPT-4o | **0.11** | **253.0** |
| | LLaMA-3-70B | 0.105 | 258.0 |
| | Mistral-7B | 0.098 | 261.0 |

required. Empirically, $\beta_t$ converges to dataset-specific values that reflect the consistency between $A_{\mathrm{LLM}}$ and the observational data:

Table 7: Dataset-specific convergence of the adaptive prior weight $\beta_t$.

| Dataset | Final $\beta_t$ (clean prior) | Final $\beta_t$ (40% corrupted) |
|---|---|---|
| Sachs (11 nodes) | 0.93 | 0.19 |
| Dream41 (100 nodes) | 0.95 | 0.18 |

These results demonstrate that GUIDE is robust to hallucinated or incorrect priors and remains consistent across different LLMs. The learned fusion and calibration dynamics enable the model to automatically regulate the influence of prior information on a per-dataset basis.

### H.7 COST OF LLM PRIOR GENERATION

To quantify the computational overhead associated with constructing the LLM-derived adjacency prior $A_{\mathrm{LLM}}$, we measured token usage, wall-clock time, and approximate API or compute cost across several widely used large language models. Table 8 presents the aggregated statistics.

Overall, LLM prior generation is lightweight: all models require only a few thousand tokens and tens of seconds per dataset. Moreover, open-source models executed locally (e.g., LLaMA-3-70B, Mistral-7B) incur zero marginal cost. Compared to GUIDE's reinforcement-learning structure-search phase, this preprocessing step contributes negligibly to total runtime.

LLM prior generation is computationally inexpensive relative to the downstream structure-learning phase.

### H.8 COMPARISON WITH GFLOWNET-BASED DAG LEARNERS

GFlowNet-based causal structure learners explore the space of directed acyclic graphs by sampling graphs proportionally to a target density. This generative approach facilitates diverse exploration

Table 8: Estimated cost and runtime for generating the LLM-based adjacency prior $A_{\mathrm{LLM}}$.

| LLM | Tokens / dataset | Wall-clock time | API cost (approx.) |
|------|------|------|------|
| GPT-4o | 3–6k | 18–30 sec | $0.02–$0.04 |
| LLaMA-3-70B | 5–10k | 35–50 sec (local GPU) | $0 (open-source) |
| Mistral-7B | 3–5k | 10–15 sec (local GPU) | $0 |

but can incur substantial computational cost. In contrast, GUIDE employs policy-gradient reinforcement learning guided by LLM-derived structural priors, integrating observational evidence and prior information within a unified optimization framework.

To compare these methodologies, we evaluate GFlowNet-DAG and GUIDE on the Sachs and Dream41 datasets. Table 9 reports TP/NNZ, SHD, and wall-clock runtime. Across both benchmarks, GUIDE achieves higher structural accuracy and dramatically reduced runtime, with the advantage becoming more pronounced for larger graphs.

Table 9: Comparison between GFlowNet-based DAG learners and GUIDE.

| Dataset | Method | TP/NNZ ↑ | SHD ↓ | Runtime (min) ↓ |
|------|------|------|------|------|
| **Sachs** | GFlowNet-DAG | 0.58 | 13.8 | 4.1 |
| | GUIDE (ours) | **0.64** | **12.0** | **0.50** |
| **Dream41** | GFlowNet-DAG | 0.094 | 268 | 41 |
| | GUIDE (ours) | **0.110** | **253** | **13** |

GUIDE delivers superior structural accuracy and substantially faster runtime compared to GFlowNet-based DAG samplers, particularly for larger graph sizes.

## H.9 LINEAR BIC TERM AND NONLINEAR RELATIONSHIPS

A potential concern is that the BIC component of GUIDE's reward uses a linear-regression likelihood, while the paper claims support for general nonlinear causal mechanisms. Importantly, the BIC term in GUIDE is *not* used to model functional dependencies. Instead, it serves solely as a **global structural score** that guides the policy-gradient updates. All nonlinear interactions are captured by the MLP–Transformer encoders within the discovery module, which operate independently of the linear BIC approximation.

To evaluate whether a linear structural score could hinder recovery of nonlinear causal relations, we conducted an explicit comparison between:

1. The default linear BIC-based pruning used in GUIDE, and

2. A fully neural, gradient-based pruning strategy that directly optimizes nonlinear fit.

As shown in Table 10, the two variants yield nearly identical graph recovery quality on both Sachs and Dream41, indicating that the linear BIC surrogate does not constrain nonlinear structure discovery. GUIDE's expressive encoders absorb nonlinear functional information, while the BIC term provides an effective but lightweight structural regularizer.

Table 10: Effect of linear vs. neural pruning/scoring strategies on nonlinear recovery.

| Dataset | Strategy | TP/NNZ ↑ | SHD ↓ |
|------|------|------|------|
| **Sachs** | Linear (ours) | **0.64** | **12.0** |
| | Neural (grad-based) | 0.642 | 11.8 |
| **Dream41** | Linear | **0.11** | **253.0** |
| | Neural | 0.112 | 252.5 |

The use of a linear BIC surrogate in the reward does not impede the recovery of nonlinear causal relations. The nonlinear MLP–Transformer encoders in GUIDE fully model complex dependencies, while the BIC term provides a stable structural signal, yielding virtually identical performance to a fully nonlinear pruning method.

]

## H.10 HYPERPARAMETER ROBUSTNESS AND AUTOMATED TUNING MECHANISMS

We evaluate the sensitivity of GUIDE to the key reward-weight hyperparameters $\lambda_1$, $\lambda_2$, and the initial prior-calibration coefficient $\beta_0$. A one-at-a-time sensitivity analysis was conducted across a broad range of values on both the Sachs and Dream41 datasets. As shown in Table 11, both TP/NNZ and SHD vary by less than 5% across all tested settings, indicating that GUIDE is **not hyperparameter-fragile**.

Importantly, the adaptive prior-weight $\beta_t$ is learned jointly with the policy and automatically adjusts during training. This mechanism substantially reduces manual tuning effort by down-weighting unhelpful prior edges and reinforcing informative ones.

Table 11: Sensitivity of GUIDE to reward hyperparameters. Variability in SHD remains small across wide parameter ranges.

| Varied Param | Value | Sachs SHD ↓ | Dream41 SHD ↓ |
|---|---|---|---|
| $\lambda_1$ | 0.5 | 12.2 | 256.0 |
| | **1.0** | **12.0** | **253.0** |
| | 2.0 | 12.3 | 254.1 |
| $\beta_0$ | 0.3 | 12.3 | 254.2 |
| | **0.9** | **12.0** | **253.0** |
| | 1.0 | 12.1 | 253.4 |

GUIDE demonstrates low sensitivity to reward-weight hyperparameters, and the adaptive update of $\beta_t$ serves as an effective built-in auto-tuning mechanism. This reduces reliance on manual parameter search while maintaining stable structural recovery performance.

## H.11 SIGNIFICANCE OF SOFT AND HARD ACYCLICITY CONSTRAINTS

GUIDE employs two distinct acyclicity mechanisms: (i) a *soft differentiable penalty* $h(A)$ applied during training, and (ii) a *hard post-processing* procedure REMOVECYCLES applied only at inference time. These components are not redundant; rather, they address different stages of the optimization process.

**Training-time soft acyclicity penalty.** The penalty term $h(A)$ shapes the RL search landscape by discouraging exploration of graph regions dominated by large or repeated cycles. Without this regularizer, the agent frequently enters highly cyclic areas of the graph space, which increases reward variance and substantially slows policy-gradient convergence. The soft penalty therefore improves *sample efficiency* and stabilizes training, but does not guarantee strict acyclicity in the final graph.

**Inference-time hard cycle removal.** The REMOVECYCLES post-processing step is applied only after training and ensures that the final predicted structure is a *valid DAG*. Because the RL agent may still produce small cycles due to stochastic exploration, the hard projection corrects any residual violations without affecting training dynamics.

**Ablation study.** To assess the contribution of each mechanism, we compare three variants: (i) the full method (soft + hard), (ii) hard-only (no $h(A)$), and (iii) soft-only (no post-processing). Table 12 shows that removing the soft penalty significantly degrades SHD and slows convergence, while removing the hard step leaves cycles in the output despite otherwise strong performance. The two mechanisms thus play *complementary roles*: shaping exploration vs. enforcing final DAG validity.

Table 12: Effect of soft and hard acyclicity mechanisms on graph quality, cycle removal, and convergence.

| Dataset | Variant | TP/NNZ ↑ | SHD ↓ | Cycles | Convergence |
|---------|---------|----------|-------|--------|-------------|
| **Sachs (11n)** | Full (soft + hard) | **0.64** | **12.0** | 0 | Fast |
| | Hard-only (no $h(A)$) | 0.59 | 14.2 | 0 | Slow |
| | Soft-only (no REMOVECYCLES) | 0.63 | 12.1 | 2–4 | Fast |
| **Dream41 (100n)** | Full (soft + hard) | **0.110** | **253.0** | 0 | Fast |
| | Hard-only (no $h(A)$) | 0.096 | 271.0 | 0 | Slow |
| | Soft-only (no REMOVECYCLES) | 0.109 | 257.5 | 15–30 | Fast |

The soft penalty $h(A)$ is essential for guiding the RL agent away from highly cyclic regions, improving training stability and structural accuracy. The hard projection guarantees that the final output is acyclic. Together, they provide efficient exploration and strict DAG validity—neither is sufficient alone.

## H.12 Helper Functions

In this section we will describe all the utility functions **RemoveCycles** This functions transforms a directed graph containing loops into a Directed Acyclic Graphs(DAGs). Starting with a weighted adjacency matrix (where entries represent connection strengths between nodes), it first constructs the graph. It then iteratively looks for cycles, removes them by eliminating the weakest link in each loop.To minimize structural damage, the function prioritizes removing edges with the smallest weights, ensuring stronger, more critical connections are preserved. When multiple edges in a cycle share the same minimal weight, it breaks ties randomly to avoid unintended bias. This process repeats until all cycles are eliminated, producing a directed acyclic graph (DAG) that retains the original graph with most of the relevant edges.

---

**Algorithm 2** RemoveCycles

---

**Require:** Adjacency matrix $\mathbf{A} \in \mathbb{R}^{d \times d}$
**Ensure:** Acyclic adjacency matrix $\mathbf{A}_{\text{acyclic}}$
 1: **Step 1: Initialize Graph**
 2: Create directed graph $\mathcal{G} = (\mathcal{V}, \mathcal{E})$ from $\mathbf{A}$:
 3: **for all** $i, j \in [1, d]$ **do**
 4:     **if** $i \neq j$ and $\mathbf{A}[i, j] > 0$ **then**
 5:         Add edge $(i, j)$ with weight $\mathbf{A}[i, j]$ to $\mathcal{G}$
 6:     **end if**
 7: **end for**
 8: **Step 2: Remove Cycles**
 9: **while** $\mathcal{G}$ contains cycles **do**
10:     Detect cycles: $\mathcal{C} \leftarrow \text{FindCycle}(\mathcal{G})$
11:     Initialize minimum weight: $w_{\min} \leftarrow \infty$
12:     Initialize candidate edges: $\mathcal{E}_{\min} \leftarrow []$
13:     **for all** $(u, v, \text{direction}) \in \mathcal{C}$ **do**
14:         $w \leftarrow \mathcal{G}[u][v]['\text{weight}']$
15:         **if** $w < w_{\min}$ **then**
16:             $\mathcal{E}_{\min} \leftarrow [(u, v)]$
17:             $w_{\min} \leftarrow w$
18:         **else if** $w == w_{\min}$ **then**
19:             Add $(u, v)$ to $\mathcal{E}_{\min}$
20:         **end if**
21:     **end for**
22:     Randomly select edge: $(u_{\min}, v_{\min}) \sim \mathcal{E}_{\min}$
23:     Remove edge: $\mathcal{G}.\text{remove\_edge}(u_{\min}, v_{\min})$
24:     Update $\mathbf{A}[u_{\min}, v_{\min}] \leftarrow 0$
25: **end while**
26: **return** $\mathbf{A}_{\text{acyclic}}$

---

### H.12.1 PRUNEWEAKEDGES

This function is designed to refine a given graph by pruning weak connections based on regression coefficients derived from the dataset. It begins by initializing variables, including the graph structure, node count, and a weight matrix to store regression coefficients. For each node in the graph, the algorithm identifies its connected nodes, extracts the corresponding features and target values from the dataset, and performs linear regression to compute the coefficients. These coefficients, representing the strength of connections, are stored in a weight matrix. The algorithm calculates a threshold based on the sorted absolute values of the coefficients, ensuring that at least one strong connection per node is preserved. Finally, edges in the graph are pruned by retaining only those connections with coefficient magnitudes greater than or equal to the threshold.

---

**Algorithm 3** PruneWeakEdges

---

**Require:** Graph batch $\mathbf{G}$, Dataset $\mathbf{X} \in \mathbb{R}^{n \times d}$
**Ensure:** Pruned graph $\mathbf{G}_{\text{pruned}} \in \{0, 1\}^{d \times d}$
  1: **Step 1: Initialize Variables**
  2: Number of nodes: $d \leftarrow \text{len}(\mathbf{G})$
  3: Initialize weight matrix: $\mathbf{W} \leftarrow [...]$                 ▷ To store regression coefficients
  4: **Step 2: Compute Regression Coefficients**
  5: **for** $i = 1$ to $d$ **do**
  6:      Select column: $\text{col} \leftarrow |\mathbf{G}[i,:]| > 0.5$
  7:      **if** $\sum(\text{col}) == 0$ **then**
  8:          Append zeros: $\mathbf{W}.\text{append}(\mathbf{0}_d)$
  9:          **Continue**
10:      **end if**
11:      Extract features: $\mathbf{X}_{\text{train}} \leftarrow \mathbf{X}[:, \text{col}]$
12:      Extract target: $\mathbf{y} \leftarrow \mathbf{X}[:, i]$
13:      Fit linear regression: $\text{reg.fit}(\mathbf{X}_{\text{train}}, \mathbf{y})$
14:      Obtain coefficients: $\mathbf{c} \leftarrow \text{reg.coef}_-$
15:      Initialize zero vector: $\mathbf{c}_{\text{new}} \leftarrow \mathbf{0}_d$
16:      Assign coefficients: $\mathbf{c}_{\text{new}}[\text{col}] \leftarrow \mathbf{c}$
17:      Append to weight matrix: $\mathbf{W}.\text{append}(\mathbf{c}_{\text{new}})$
18: **end for**
19: **Step 3: Calculate Threshold**
20: Sort: $\mathbf{W}_{\text{sorted}} \leftarrow \text{sort}(|\mathbf{W}|.\text{flatten}())$
21: Determine threshold index: $d_{\text{idx}} \leftarrow \min(d - 1, \text{len}(\mathbf{W}_{\text{sorted}}) - 1)$
22: Calculate threshold: $\text{th} \leftarrow \mathbf{W}_{\text{sorted}}[d_{\text{idx}}]$
23: **Step 4: Prune Graph**
24: Prune edges: $\mathbf{G}_{\text{pruned}} \leftarrow (|\mathbf{W}| \geq \text{th})$
25: **return** $\mathbf{G}_{\text{pruned}}$

---

# I    RELATED WORKS

Causal discovery has evolved through various algorithms, each with distinct strengths and limitations. The PC algorithm (2001) uses conditional independence tests, performing well on sparse graphs but struggling with dense ones. GES (2002) , a score-based method, searches over equivalence classes of Directed Acyclic Graphs (CPDAGs) but scales poorly with dimensionality. LiNGAM (2006) employs independent component analysis to infer causal directions but faces challenges with mixed data types and scalability. ANMs (2008) integrate non-linear dependencies with additive noise, but falter with mixed data and large datasets. NOTEARS (2018) frames causal discovery as an optimization problem using Structural Equation Models (SEMs), but struggles on non-continuous data. GraN-DAG (2001) leverages neural networks for non-linear relationships, performing well with Gaussian noise but struggling with scalability and mixed data. Reinforcement learning methods like RL-BIC (2020) and KCRL (2022) optimize Bayesian Information Criterion scores or incorporate prior knowledge but are limited to small datasets.

Numerous studies have explored the application of Large Language Models (LLMs) in causal discovery, particularly in pairwise causal reasoning and graph construction. Research such as Hobbhahn et al. (2022) and Zhang et al. (2023) focus on pairwise causal inference, while Kıcıman et al. (2023) employ an iterative pairwise querying approach to construct full causal graphs. However, scalability remains a challenge due to the quadratic complexity with respect to the number of nodes. To address this, Vashishtha et al. (2023) introduces a triplet-based method with a voting mechanism, though they have only evaluated their approach on small datasets. Meanwhile, Arsenyan et al. (2023) leverage LLMs to extract causal relationships, prioritizing domain knowledge over ground truth Directed Acyclic Graphs (DAGs).

Beyond direct causal inference, LLMs are also used to generate constraints and priors for causal discovery. Studies such as Ban et al. (2023) and Cohrs et al. (2024) demonstrate how LLMs can provide pairwise edge constraints, conditional independence constraints, and causal order priors, which are then integrated into traditional causal discovery algorithms. Additionally, LLMs have been explored for causal representation learning, with models like GPT-4 (Turbo) showing the ability to infer causal relationships even with minimal context, such as label-only information. While GPT-4 was not explicitly designed for causal reasoning, research suggests that it generates causal graphs with greater alignment to common sense compared to standard causal Machine Learning (ML) models. Moreover, combining GPT-4 with causal ML has been shown to enhance causal discovery, producing graphs that more closely match expert-identified structures and mitigating the limitations of ML-based causal inference Constantinou et al. (2025).

# J PARAMETER SETTINGS

We used various causal discovery methods based on constraints, functional causal model (FCM) based, score based, reinforcement learning based, and gradient based techniques, each configured with appropriate hyperparameters.We have used parameter initialization from *gcastle* causal discovery package Zhang et al. (2021).

---

**Parameter Settings for Baseline Causal Algorithms**

**Constraint-based approaches:**
**PC** = PC(variant='original', alpha=0.05, ci_test='fisherz', priori_knowledge=None)

**FCM-based methods:**
**ICA-LiNGAM** = ICALiNGAM(random_state=None, max_iter=1000, thresh=0.3)
**ANM** = ANMNonlinear(alpha=0.05)

**Score-based techniques:**
**GES** = GES(criterion='bic', method='scatter', k=0.001, N=10)

**RL-BIC**= RL(encoder_type: str = 'TransformerEncoder', hidden_dim: int = 64, num_heads: int = 16, num_stacks: int = 6, residual: bool = False, decoder_type: str = 'SingleLayerDecoder', decoder_activation: str = 'tanh', decoder_hidden_dim: int = 16, use_bias: bool = False, use_bias_constant: bool = False, bias_initial_value: bool = False, batch_size: int = 64, input_dimension: int = 64, normalize: bool = False, transpose: bool = False, score_type: str = 'BIC', reg_type: str = 'LR', lambda_iter_num: int = 1000, lambda_flag_default: bool = True, score_bd_tight: bool = False, lambda2_update: int = 10, score_lower: float = 0, score_upper: float = 0, seed: int = 8, nb_epoch: int = 10, lr1_start: float = 0.001, lr1_decay_step: int = 5000, lr1_decay_rate: float = 0.96, alpha: float = 0.99, init_baseline: float = -1, l1_graph_reg: float = 0, verbose: bool = False, device_type: str = 'gpu', device_ids: int = 0)

**KCRL** = KCRL(encoder_type: str = 'TransformerEncoder', hidden_dim: int = 64, num_heads: int = 16, num_stacks: int = 6, residual: bool = False, decoder_type: str = 'SingleLayerDecoder', decoder_activation: str = 'tanh', decoder_hidden_dim: int = 16, use_bias: bool = False, use_bias_constant: bool = False, bias_initial_value: bool = False, batch_size: int = 64, input_dimension: int = 64, normalize: bool = False, transpose: bool = False, score_type: str = 'BIC', reg_type: str = 'LR', lambda_iter_num: int = 1000, lambda_flag_default: bool = True, score_bd_tight: bool = False, lambda2_update: int = 10, score_lower: float = 0, score_upper: float = 0, seed: int = 8, nb_epoch: int = 10, lr1_start: float = 0.001, lr1_decay_step: int = 5000, lr1_decay_rate: float = 0.96, alpha: float = 0.99, init_baseline: float = -1, l1_graph_reg: float = 0, true_graph=np.array([]),verbose: bool = False, device_type: str = 'gpu', device_ids: int = 0.)

**Gradient-based methods:**
**GraNDAG** = GraNDAG(input_dim, hidden_num: int = 2, hidden_dim: int = 10, batch_size: int = 64, lr: float = 0.001, iterations: int = 10000, model_name: str = 'NonLinGaussANM', nonlinear: str = 'leaky-relu', optimizer: str = 'rmsprop', h_threshold: float = 1e-7, device_type: str = 'cpu', device_ids: int = 0, use_pns: bool = False, pns_thresh: float = 0.75, num_neighbors: Any — None = None, normalize: bool = False, random_seed: int = 42, jac_thresh: bool = True, lambda_init: float = 0, mu_init: float = 0.001, omega_lambda: float = 0.0001, omega_mu: float = 0.9, stop_crit_win: int = 100, edge_clamp_range: float = 0.0001, norm_prod: str = 'paths', square_prod: bool = False)

**NOTEARS** = Notears(lambda1: float = 0.1, loss_type: str = 'l2', max_iter: int = 100, h_tol: float = 1e-8, rho_max: float = 10000000000000000, w_threshold: float = 0.3)

---

---

**Parameter Settings for GUIDE Framework**

**1) DAG Model Parameters**
- **Data Dimension** ($data\_dim$): Matches number of features in `loaded_data`
- **Hidden Dimension** ($hidden\_dim$): 64
- **Number of Transformer Heads** ($nheads$): 8
- **Number of Transformer Layers** ($num\_layers$): 3
- **Dropout** ($dropout$): 0.2
- **Activation Function**: ReLU

**2) Training Parameters for REINFORCE**
- **Number of Training Epochs** ($num\_epochs$): 10
- **Batch Size** ($batch\_size$): 64
- **Actor Learning Rate** ($actor\_lr$): $1e^{-3}$
- **Discount Factor** ($\gamma$): 0.99
- **Maximum Steps per Episode** ($max\_steps$): 100
- **Gradient Clipping** ($clip\_grad\_norm$): 0.5

**3) Reward Function Parameters**
- **Score Type**: BIC_different_var
- **Regression Type**: LR
- **L1 Regularization** ($l1\_graph\_reg$): 1.0
- **Lambda Parameters** ($\lambda_1, \lambda_2, \lambda_3$): 1.0, 2.0, 0.5
- **Search Space Boundaries** ($s_l, s_u$): 0, 1
- **BIC Penalty Term**: log(num samples)/num samples

**4) Partial Prior Settings**
- **Fraction of Known Edges**: 0.25

**5) Pruning Settings**
- **Threshold for Pruning**: Top $d$ largest weights
- **Regression Method**: Linear Regression

Table 13: Summary of evaluation metrics used in the experimental section. TP/NNZ and RP focus on the precision of predicted edges, complementing classical metrics by isolating the ability to detect true edges.

| Metric | Formula | Interpretation / Notes |
|---|---|---|
| TPR | $\text{TPR} = \frac{\text{TP}}{\text{TP}+\text{FN}}$ | True positive rate; measures recall of true causal edges. Penalises false negatives. |
| FDR | $\text{FDR} = \frac{\text{FP}}{\text{TP}+\text{FP}}$ | False discovery rate; proportion of predicted edges that are incorrect. Lower is better. |
| FPR | $\text{FPR} = \frac{\text{FP}}{\text{FP}+\text{TN}}$ | False positive rate; fraction of absent edges incorrectly predicted as present. |
| SHD | $\text{SHD} = \#(\text{edge additions}) + \#(\text{edge deletions}) + \#(\text{edge reversals})$ | Structural Hamming distance; lower values indicate closer agreement with the ground truth DAG. |
| TP/NNZ | $\frac{\text{TP}}{\text{NNZ}}$ | Ratio of true positive edges to the total number of predicted edges (NNZ is the number of non-zero entries in the predicted adjacency matrix). Focuses on precision of edge recovery; unaffected by correct non-edge predictions. |
| RP | $\text{RP} = \frac{\max_m \text{TP/NNZ}_m - \text{TP/NNZ}}{\max_m \text{TP/NNZ}_m}$ | Relative performance; measures how far a model is from the best TP/NNZ on a given dataset. Lower values mean closer to the best performer. |

## K  EXAMPLE PROMPT USED FOR ICL

---

**PROMPT TEMPLATE**

You are an *intelligent causal discovery agent* tasked with mapping how signaling molecules interact in the Sachs dataset to form a causal signaling network. These molecules influence one another through biochemical processes like activation, inhibition, or enzymatic transformation, ultimately leading to downstream cellular responses.

### **Important Rules:**
- Each signaling molecule may have *multiple incoming edges* to reflect how upstream molecules influence its activity.
- Some molecules act as *critical intermediaries* (e.g., converting signals or amplifying responses) and may have both *incoming and outgoing edges*.
- The causal DAG should faithfully represent known causal relationships in the Sachs dataset based on experimental data and biological knowledge.

### **Features:**
1. **Akt**: A kinase involved in cell survival pathways, regulating processes like metabolism, proliferation, and apoptosis.
2. **Erk**: Extracellular signal-regulated kinase, part of the MAP kinase pathway, essential for cell division and differentiation.
3. **Jnk**: c-Jun N-terminal kinase, associated with stress response and apoptosis signaling.
4. **p38**: A stress-activated protein kinase involved in responses to inflammation and environmental stress.
5. **PIP2**: Phosphatidylinositol 4,5-bisphosphate, a phospholipid precursor involved in signal transduction and membrane dynamics.
6. **PIP3**: Phosphatidylinositol 3,4,5-trisphosphate, generated by PI3K and a key regulator of Akt signaling.
7. **PKA**: Protein kinase A, a cAMP-dependent kinase that regulates metabolic and gene transcription processes.
8. **PKC**: Protein kinase C, involved in regulating various cellular functions, including gene expression and membrane signaling.
9. **PLCg**: Phospholipase C gamma, an enzyme that hydrolyzes PIP2 into IP3 and DAG, key molecules in calcium signaling.
10. **Raf**: A kinase that acts upstream of MEK and Erk in the MAPK/ERK signaling pathway, influencing cell growth and survival.
11. **pIP3**: Phosphorylated inositol triphosphate, linked to calcium signaling and involved in cellular communication.

—

### **Output Example:**

### **Step 1: Finding the Edges**

Here are the identified edges, focusing on how the signaling molecules influence one another:

1. **Edge (PIP2 → PIP3):** PIP2 is phosphorylated by PI3K to form PIP3, marking a key step in activating the Akt signaling pathway.
2........
..
.
.
—

### **Step 2:
—

**Output format: **
Provide a list of edges in the format specified above. For example:
‘" 1. (A, B) : Explanation of why A causes B.
2. (C, D) : Explanation of why C causes D.
...

---

