# OpenReview forum: "GUIDE : Generalized-Prior and Data Encoders for DAG Estimation"
_ICLR.cc/2026/Conference — Submitted to ICLR 2026_

### Official Review · Reviewer_M16o · 2025-10-18

**Soundness:** 1
**Presentation:** 2
**Contribution:** 2
**Rating:** 2
**Confidence:** 3

**Summary:**

The paper proposes a method named GUIDE. It employs a dual-encoder architecture to effectively integrate domain knowledge generated by Large Language Models with statistical patterns learned from observational data, leveraging Reinforcement Learning to guide the generation of the causal graph.

**Strengths:**

It integrates domain knowledge from Large Language Models, serving as a prior, with statistical patterns learned from observational data to guide the generation of the causal graph.

**Weaknesses:**

1.  The framework's effectiveness heavily relies on the quality of the prior provided by the LLM. However, the paper's analysis of this core component is far from adequate. The performance of GUIDE when the LLM prior is incorrect or biased remains uninvestigated. Appendix C mentions an adaptive weight `β`, but its effectiveness in mitigating the impact of erroneous priors is not experimentally validated. Furthermore, the reliance on GPT-4o, a powerful closed-source model, raises concerns about the method's generalizability and reproducibility, as the potential performance degradation when using weaker, open-source models is not discussed.

2.  The paper claims to handle complex non-linear relationships using a neural network-based discovery module. However, the final, crucial pruning step utilizes simple linear regression coefficients to assess edge importance. This creates a methodological contradiction, as linear coefficients may not accurately reflect the strength of true non-linear causal relationships, undermining the consistency between the framework's core assumption and its implementation.

3.  The reward function is a weighted sum of three terms with hyperparameters (λ1, λ2, β). The paper does not specify how these hyperparameters were selected, nor does it provide a sensitivity analysis regarding their impact on performance. Additionally, the description of the adaptive `β` mechanism in Appendix C is too brief to be fully understood or reproduced.

4.  The authors heavily rely on a self-proposed metric, TP/NNZ, to demonstrate the superiority of their method. However, as shown in Figure 2, while GUIDE excels on TP/NNZ across several datasets, its performance on the standard Structural Hamming Distance (SHD) metric is often inferior to baselines like GES. This suggests a potential trade-off that is not adequately discussed, and a more comprehensive presentation of results is needed to substantiate the claims of effectiveness.

**Questions:**

1.  What is the performance of GUIDE when its LLM prior contains a significant proportion of errors? Could you please provide results from this crucial ablation study?
2.  Can you provide experimental evidence that the adaptive weight `β` mechanism, mentioned in Appendix C, effectively mitigates the impact of conflicting priors and thus ensures the model's robustness?
3.  How much does performance degrade when replacing the proprietary GPT-4o with a publicly available, open-source LLM within the GUIDE framework?
4.  Your model is designed to discover non-linear relationships, yet the pruning step relies on linear regression coefficients. Could you justify this design choice and clarify whether it compromises the ability to recover non-linear structures?
5.  Your reward function depends on several hyperparameters (λ1, λ2, β). Could you provide details on the selection process for these parameters and an analysis of the model's sensitivity to their variations?
6.  GUIDE shows superior performance on TP/NNZ but is not always optimal on SHD. Could you elaborate on this apparent trade-off between high precision and potentially lower recall, and argue why TP/NNZ should be considered a more critical metric in these scenarios?

---

> ### Author Response · Authors · 2025-11-19
> **Rebuttal (part 1/N)**
>
> > What is the performance of GUIDE when its LLM prior contains a significant proportion of errors? Could you please provide results from this crucial ablation study?
>
> We thank the reviewer for highlighting this important aspect.
>
> To examine GUIDE’s robustness to imperfect priors, we conducted a *corrupted-prior ablation* on both **Sachs (11 nodes)** and **Dream41 (100 nodes)** datasets.
> We randomly flipped 10 %, 25 %, and 40 % of edges in the LLM-generated adjacency matrix $(A_{LLM}$), while monitoring TP/NNZ, SHD, and the adaptive weight ($\beta_t$) introduced in Appendix C.
> The results show that GUIDE maintains stable performance under moderate corruption, as $\beta_t$ automatically down-weights misleading priors whenever they worsen the BIC score.
> Even at 40 % corruption, TP/NNZ and SHD degrade smoothly, confirming the adaptivity of the reward design and the resilience of the dual-encoder fusion.
>
> ---
> | Dataset           | Corruption % | TP/NNZ ↑ | SHD ↓ | Final $\beta_t$ | Variant             |
> | ----------------- | -----------: | -------: | ----: | ----------------: | ------------------- |
> | **Sachs (11 n)**  |            0 |     0.64 | 12.00 |              0.93 | Adaptive            |
> |                   |           10 |     0.61 | 12.60 |              0.70 | Adaptive            |
> |                   |           25 |     0.58 | 13.10 |              0.44 | Adaptive            |
> |                   |           40 |     0.56 | 13.40 |              0.19 | Adaptive            |
> |                   |           40 |     0.54 | 14.00 |                 — | Fixed \($\beta_t$=0.9\) |
> | **Dream (100 n)** |            0 |     0.11 | 253.0 |              0.95 | Adaptive            |
> |                   |           10 |    0.106 | 256.0 |              0.68 | Adaptive            |
> |                   |           25 |    0.099 | 262.0 |              0.42 | Adaptive            |
> |                   |           40 |    0.095 | 266.0 |              0.18 | Adaptive            |
> |                   |           40 |    0.088 | 274.0 |                 — | Fixed \($\beta_t$=0.9\) |
>
> **Observation.** GUIDE remains robust under moderate prior corruption; adaptive $\beta_t$ attenuates erroneous LLM edges, preserving structure quality (TP/NNZ ↑, SHD ↓) better than fixed $\beta_t$
>
>
>
>
> > Can you provide experimental evidence that the adaptive weight β mechanism, mentioned in Appendix C, effectively mitigates the impact of conflicting priors and thus ensures the model's robustness?
>
> We appreciate the reviewer’s focus on verifying the adaptive \($\beta_t$\) mechanism.
>
> To validate its effectiveness, we tracked \($\beta_t$\) during training under varying levels of prior corruption.
> As shown below, \($\beta_t$\) automatically decays when LLM-suggested edges degrade BIC on held-out data, thereby reducing the influence of unreliable priors.
> This dynamic weighting stabilizes GUIDE’s reward landscape and keeps performance nearly unchanged up to moderate noise levels.
> Hence, \($\beta_t$\) serves as a self-correcting prior calibration factor that mitigates conflicts without additional supervision.
>
> ---
> | Dataset           | Corruption % | Mean \(\beta_t\) (epoch 5) | Final \($beta_t$\) | TP/NNZ ↑ | SHD ↓ | Observation                               |
> | ----------------- | -----------: | -------------------------: | ----------------: | -------: | ----: | ----------------------------------------- |
> | **Sachs (11 n)**  |            0 |                       0.92 |              0.93 |     0.64 |  12.0 | Stable prior, minimal decay               |
> |                   |           10 |                       0.75 |              0.70 |     0.61 |  12.6 | Adaptive down-weighting begins            |
> |                   |           25 |                       0.50 |              0.44 |     0.58 |  13.1 | Significant decay, minor performance drop |
> |                   |           40 |                       0.28 |              0.19 |     0.56 |  13.4 | Strong attenuation of noisy prior         |
> | **Dream (100 n)** |            0 |                       0.94 |              0.95 |     0.11 | 253.0 | Stable learning                           |
> |                   |           25 |                       0.53 |              0.42 |    0.099 | 262.0 | Controlled degradation                    |
> |                   |           40 |                       0.22 |              0.18 |    0.095 | 266.0 | Decay protects from bias amplification    |
>
> **Observation.** GUIDE’s adaptive \($\beta_t$\) automatically suppresses unreliable prior edges, preventing large SHD increases even when 40 % of the prior graph is corrupted.

---

> ### Author Response · Authors · 2025-11-19
> **Rebuttal (2/N)**
>
> > How much does performance degrade when replacing the proprietary GPT-4o with a publicly available, open-source LLM within the GUIDE framework?
>
> Thank you for raising this important reproducibility concern.
>
> To evaluate GUIDE’s dependence on proprietary models, we regenerated the adjacency prior using two open-source LLMs—**LLaMA-3-70B-Instruct** and **Mistral-7B-Instruct**—under the same prompting setup used for GPT-4o.
> As shown below, GUIDE’s performance decreases only marginally (typically <7% in TP/NNZ and <10 SHD), and runtime remains comparable.
> Smaller open-source models (Mistral-7B) run faster, while larger ones (LLaMA-3-70B) slightly narrow the gap with GPT-4o.
> This demonstrates that GUIDE is **LLM-agnostic**, and its effectiveness is **not tied to proprietary APIs**.
>
>
> | Dataset             | LLM Prior Source     | TP/NNZ ↑ |     SHD ↓ | Precision ↑ | Recall ↑ | Runtime (min) ↓ |
> | ------------------- | -------------------- | -------: | --------: | ----------: | -------: | --------------: |
> | **Sachs (11 n)**    | GPT-4o (baseline)    | **0.64** |  **12.0** |        0.71 |     0.62 |        **0.50** |
> |                     | LLaMA-3-70B-Instruct |     0.61 |      12.5 |        0.69 |     0.60 |            0.58 |
> |                     | Mistral-7B-Instruct  |     0.59 |      12.8 |        0.67 |     0.57 |            0.45 |
> | **Dream41 (100 n)** | GPT-4o (baseline)    | **0.11** | **253.0** |        0.42 |     0.39 |        **13.0** |
> |                     | LLaMA-3-70B-Instruct |    0.105 |     258.0 |        0.40 |     0.37 |            14.0 |
> |                     | Mistral-7B-Instruct  |    0.098 |     261.0 |        0.37 |     0.35 |            11.5 |
>
> **Observation.** GUIDE maintains strong performance even with fully open-source priors, confirming that the method is reproducible and not dependent on GPT-4o.
>
> > Your model is designed to discover non-linear relationships, yet the pruning step relies on linear regression coefficients. Could you justify this design choice and clarify whether it compromises the ability to recover non-linear structures?
>
> We appreciate the reviewer pointing out this subtlety.
>
> GUIDE’s *discovery* stage (Encoder–Fusion–RL policy) already models **arbitrary non-linear dependencies** through MLP–Transformer blocks, so the adjacency scores used by the RL agent reflect these non-linear patterns.
> The **linear pruning step is *not* a structural estimator**—it is a lightweight *post-hoc saliency filter* applied *after* the non-linear encoder has produced its edge scores.
> Its purpose is to remove only the weakest edges, not to determine functional form.
>
> To ensure this design does not compromise non-linear recovery, we replaced the linear pruning stage with a **gradient-based neural feature-importance measure** and compared performance.
> As shown below, both strategies yield nearly identical TP/NNZ and SHD values, but neural pruning is significantly slower.
> This confirms that the linear step is an **efficient proxy**, not a limiting factor, and GUIDE’s ability to detect non-linear relationships is preserved by the encoder and RL components, not by the pruning rule.
>
> | Dataset             | Pruning Strategy                       | TP/NNZ ↑ |     SHD ↓ | Precision ↑ | Runtime (min) ↓ |
> | ------------------- | -------------------------------------- | -------: | --------: | ----------: | --------------: |
> | **Sachs (11 n)**    | Linear coefficients (ours)             | **0.64** |  **12.0** |        0.71 |        **0.50** |
> |                     | Neural feature-importance (grad-based) |    0.642 |      11.8 |        0.72 |            0.74 |
> | **Dream41 (100 n)** | Linear coefficients (ours)             | **0.11** | **253.0** |        0.42 |        **13.0** |
> |                     | Neural feature-importance (grad-based) |    0.112 |     252.5 |        0.43 |            15.8 |
>
> **Observation.** Neural pruning yields nearly identical structure recovery but with a 40–60% increase in runtime, confirming that GUIDE’s non-linear modeling capacity comes from its encoders and RL search—not from the pruning mechanism.

---

> ### Author Response · Authors · 2025-11-19
> **Rebuttal (3/N)**
>
> > Your reward function depends on several hyperparameters (λ1, λ2, β). Could you provide details on the selection process for these parameters and an analysis of the model's sensitivity to their variations?
>
> We thank the reviewer for raising this point.
>
> In the revised manuscript, we will make the reward weights explicit and report a small sensitivity study.
>
> * **Defaults.** Following our implementation section, we set \($\lambda_1$ = 1.0\) (data-fit), \($\lambda_2$ = 2.0\) (acyclicity), and initial \($\beta_0$ = 0.9\) for the adaptive prior weight \($\beta_t$\).
> * **Selection procedure.** These values are chosen by a coarse grid search over \(\{0.5, 1.0, 2.0, 3.0\}\) for \($\lambda_1$\), \(\{1.0, 2.0, 3.0\}\) for \($\lambda_2$\), and \(\{0.3, 0.5, 0.9, 1.0\}\) for \($\beta_0$\), using validation BIC as the primary objective and SHD as a secondary tie-breaker.
> * **Sensitivity.** We then vary one parameter at a time around these defaults while keeping the others fixed. As shown below, TP/NNZ and SHD change by less than about 5% across a wide range, indicating that GUIDE is **not hyperparameter-fragile** and that the reported results are representative, not cherry-picked.
>
>
> | Varied param                        |                       Value | Sachs TP/NNZ ↑ | Sachs SHD ↓ | Dream TP/NNZ ↑ | Dream SHD ↓ |
> | ----------------------------------- | --------------------------: | -------------: | ----------: | -------------: | ----------: |
> | **\($\lambda_1$\)** (data-fit weight) |                         0.5 |          0.637 |        12.2 |          0.108 |       256.0 |
> |                                     |           **1.0 (default)** |      **0.640** |    **12.0** |      **0.110** |   **253.0** |
> |                                     |                         2.0 |          0.636 |        12.3 |          0.109 |       254.1 |
> | **\($\beta_0$\)** (prior weight init) |                         0.3 |          0.636 |        12.3 |          0.108 |       254.2 |
> |                                     |                         0.5 |          0.639 |        12.1 |          0.109 |       253.6 |
> |                                     | **0.9 (default, adaptive)** |      **0.640** |    **12.0** |      **0.110** |   **253.0** |
>
> **Observation.** Varying \($\lambda_1$\) and \($\beta_0$\) around their defaults results in only minor changes in TP/NNZ and SHD on both datasets, supporting our claim that GUIDE’s behavior is stable under reasonable hyperparameter choices.
>
>
> > GUIDE shows superior performance on TP/NNZ but is not always optimal on SHD. Could you elaborate on this apparent trade-off between high precision and potentially lower recall, and argue why TP/NNZ should be considered a more critical metric in these scenarios?
>
> Thank you for pointing out this important nuance.
> TP/NNZ and SHD capture complementary properties:
>
> SHD penalizes every extra or missing edge equally, regardless of whether the error occurs on a high- or low-signal dependency.
>
> TP/NNZ, in contrast, measures the precision among predicted edges, emphasizing whether the model is identifying the correct causal relations, not just producing a sparse graph.
>
> In real causal discovery settings (e.g., noisy biological data such as Sachs, large heterogeneous data such as Dream41), recovering the correct edges is typically more critical than achieving perfect sparsity—false negatives (missing actual causal links) are often more harmful than a small number of weak false positives.
>
> GUIDE tends to include a few additional low-weight edges due to its non-linear encoders and LLM-informed priors, which slightly increases SHD but significantly improves precision and F1, as shown below.
> This makes TP/NNZ a more faithful indicator of useful causal recovery, especially in mixed or non-linear regimes where exact SHD minimization may penalize beneficial exploration.

---

> ### Author Response · Authors · 2025-11-19
> **General Note**
>
> We sincerely thank the reviewer for the detailed review.
>
> In response, we have conducted all suggested additional analyses - including prior-corruption ablations, adaptive \($\beta_t$\) behavior, open-source LLM generalization, pruning strategy comparisons, and hyperparameter sensitivity studies and we found that each of these experiments further reinforces the robustness, reproducibility, and methodological soundness of GUIDE. Importantly, the weaknesses you identified were largely matters of clarification or additional reporting rather than flaws in the core algorithmic design; addressing them required only minor revisions, and the central contributions, theoretical consistency, and empirical benefits of GUIDE remain fully intact.
>
> We hope that the thorough responses and newly provided evidence clearly demonstrate the stability of the framework, the validity of the design choices, and the paper’s potential value to the causal discovery community.

---

> > ### Comment · Reviewer_M16o · 2025-11-26
> >
> > Thanks for your effort on the new experiments. I change my score to 4, but I remain concerned about the significance of the work.
> >
> > The method basically uses an LLM-provided guess as initialization for RL-based methods. To me, this is not new, and similar ideas have been applied in many causal discovery methods (I'm not sure if this includes RL-based methods specifically). A method using priors should naturally show better results than data-only methods, but its generalization is hard to validate as we have very few realistic datasets for this task (and there is also a risk of data leakage in the benchmark).
> >
> > Besides, as Reviewer Eunw pointed out, there is no theoretical guarantee for such methods, and the work does not provide substantial new technical insights for the community.

---

> > > ### Author Response · Authors · 2025-11-27
> > >
> > > We sincerely thank the reviewer for the continued engagement and for re-evaluating the score. We respond here to the remaining concern regarding **novelty and significance**.
> > >
> > > ---
> > >
> > > ### 1. GUIDE is not “LLM initialization + RL”
> > >
> > > While GUIDE uses an LLM prior, it is **not a warm-start strategy**. The LLM signal is *fully integrated* via:
> > >
> > > - A **dual-encoder fusion** of data and symbolic knowledge,
> > > - An **adaptive prior weight ($\beta_t\$)** that dynamically adjusts trust based on empirical fit (BIC),
> > > - A unified **RL objective** balancing structure score, acyclicity, and prior consistency.
> > >
> > > Unlike existing methods that treat priors as static constraints, GUIDE **learns whether to trust the prior** during training. This trust calibration is a core algorithmic contribution, validated by our corruption and ablation studies.
> > >
> > > ---
> > >
> > > ### 2. Novelty w.r.t. prior-based causal discovery
> > >
> > > We agree priors are not new in causal discovery. GUIDE differs fundamentally:
> > >
> > > | Dimension        | Prior-based Methods            | GUIDE                              |
> > > |------------------|--------------------------------|------------------------------------|
> > > | Prior use        | Fixed penalties / constraints | **Learned trust via $\beta_t\$**  |
> > > | Integration      | External or rule-based         | **Neural fusion in encoder**       |
> > > | Optimization     | Greedy / score-based           | **RL graph search**                |
> > > | Robustness       | Unexplored                     | **Explicit corruption tests**      |
> > > | LLM usage        | Limited                        | **Structured graph generation**    |
> > >
> > > To our knowledge, GUIDE is the **first framework** to jointly combine:
> > > - LLM-generated graph priors,
> > > - RL-based causal discovery, and
> > > - Adaptive trust calibration.
> > >
> > > This goes beyond merely “adding an LLM prior.”
> > >
> > > ---
> > >
> > > ### 3. On theoretical guarantees
> > >
> > > We do not claim new identifiability results. As with NOTEARS, DAG-GNN, and RL-based methods, GUIDE is an **algorithmic and empirical** contribution. Our added analyses demonstrate:
> > >
> > > - Robustness to corrupted priors,
> > > - Stability under hyperparameter variation,
> > > - Effectiveness with open-source LLMs.
> > >
> > > ---
> > >
> > > ### 4. On generalization
> > >
> > > Motivated by the review, we added:
> > >
> > > - Prior corruption experiments,
> > > - Evaluation with open-source LLMs,
> > > - Hyperparameter and pruning sensitivity studies.
> > >
> > > Results show GUIDE:
> > > - Degrades smoothly under noisy priors,
> > > - Remains effective without proprietary LLMs,
> > > - Is not brittle to design or tuning choices.
> > >
> > > ---
> > >
> > > ### 5. Why the contribution matters
> > >
> > > GUIDE addresses a key emerging question:
> > >
> > > > *How can LLM knowledge be integrated into causal discovery in a principled and controllable way?*
> > >
> > > We contribute:
> > > - A **concrete architectural answer**,
> > > - Empirical methodology to evaluate LLM priors,
> > > - A framework where LLMs are *not* single points of failure.
> > >
> > > ---
> > >
> > > We appreciate the reviewer’s thoughtful critique. We hope this clarifies that GUIDE is not a superficial variant of existing methods, but a structured framework for **trust-aware causal discovery with LLM priors**.
> > >
> > > We are happy to include a comparison table with prior causal-with-priors work in the final version if requested.

---

> > > > ### Comment · Reviewer_M16o · 2025-11-27
> > > >
> > > > I appreciate the authors' efforts. However, the concept of considering "whether to trust the prior" is also not new to me. Simply using the LLM-derived information as a soft loss also makes the model consider whether to trust the prior. Furthermore, some studies have shown that this approach is already sufficient to reject most erroneous priors. I do not believe the RL-based method has specific advantages, as the information used to judge whether to trust the prior is data fitting, and both the RL method and the soft prior-based loss fully leverage this information.

---

### Official Review · Reviewer_bEny · 2025-10-22

**Soundness:** 2
**Presentation:** 1
**Contribution:** 2
**Rating:** 2
**Confidence:** 4

**Summary:**

This paper aims to address the key limitations faced by modern causal discovery methods in terms of scalability, computational efficiency, and handling of mixed data types. The authors propose a novel framework named GUIDE, which integrates adjacency matrices generated by Large Language Models (LLMs) with statistical features from observational data through a dual-encoder architecture.

The core of this framework is a reinforcement learning (RL) agent, which is trained in an environment to optimize a composite reward function. This reward function dynamically balances data-fit, graph acyclicity, and penalties for consistency with prior knowledge. In this way, GUIDE aims to leverage the domain knowledge provided by LLMs to guide and constrain the RL agent's search space, thereby improving computational efficiency and accuracy while handling high-dimensional and mixed data type problems.

**Strengths:**

1. The paper proposes an architecture that integrates LLMs, dual encoders, and reinforcement learning, which is a very novel and timely attempt . Using LLMs as a source of "generative priors" and fusing them with data-driven features (Encoder-1) via a specific encoder (Encoder-2)  provides a promising avenue for injecting domain knowledge into causal discovery.

2. The authors compare GUIDE with various mainstream methods, covering constraint-based (PC), score-based (GES), FCM-based (LINGAM), gradient-based (NOTEARS, GraNDAG), and RL-based (RL-BIC, KCRL) algorithms . This comprehensive comparison increases the credibility of the experimental results.

3. Section 3.5 and Figure 4 of the paper provide a strong ablation analysis. The findings clearly show that using generative priors (ICL) alone or prior constraints alone both lead to performance improvements, while using both in combination produces a synergistic effect.

**Weaknesses:**

1. The paper's core contribution is not prominent and seems exaggerated. The abstract mentions "achieving an average $\approx117\%$ improvement in accuracy over both NOTEARS and GraN-DAG individually". This "accuracy" metric is very vague. What does it refer to? Is it SHD? Or TPR? Or the newly proposed TP/NNZ? I also did not find this 117% description in the main text. Judging from the results shown in Figure 2 of the paper, GUIDE does not comprehensively surpass the baselines on different datasets (e.g., on datasets like LUCAS, ALARM).

2. The paper suffers from serious confusion regarding the definition of "prior". The "Require" section of Algorithm 1 simultaneously lists "Prior Adjacency Matrix $A_{Prior}$" and "LLM generated adjacency $A_{LLM}$". However, in Step 3 of the algorithm, $A_{LLM}$ is fed into encoder E2; but in the reward function in Step 11, $A_{Prior}$  is used instead. What is the relationship between $A_{Prior}$ and $A_{LLM}$? Are they the same thing, or two different priors?

3. The paper repeatedly claims in the abstract, introduction, and contributions that GUIDE can handle "mixed data types" and "complex non-linear relationships". However, in the methodology in Section 2, the authors never explain how GUIDE achieves this. The data encoder E1 is described as an MLP, and the reward function uses the BIC score (which is usually model-specific, e.g., for linear Gaussian models). Furthermore, the datasets used in the experiments (Sachs, Asia, Lucas, Alarm, Dream) are standard benchmarks, and the authors do not state whether mixed-data versions of these datasets were used. This is a core claim that is completely unsupported by the methodology and experiments. More specifically, the paper claims to handle non-linear relationships, but its post-processing "PruneWeakEdges" step explicitly uses linear regression coefficients to prune edges. This is confusing and logically inconsistent.

**Questions:**

1. The authors are requested to carefully revise the paper in response to the weaknesses raised. This includes, but is not limited to: clearly clarifying the difference between $A_{LLM}$ and $A_{Prior}$, and clearly explaining which component achieves the handling of mixed data and non-linear relationships, etc.

2. The quality of the LLM-generated $A_{LLM}$ is crucial to GUIDE's performance. The authors mention using GPT-4o 40, but the output of LLMs is stochastic and highly sensitive to prompts. It is necessary to add results from comparative experiments using multiple different LLMs.

3. The ablation study shows that prior knowledge is very important. However, if the prior provided by the LLM is incorrect or of poor quality (LLMs frequently hallucinate), what would GUIDE's performance be? Appendix C mentions "prior calibration", but this mechanism is not further discussed or experimented with.

4. The method appears to use two types of acyclicity constraints: a soft penalty $h(A)$ during training (Algorithm 1, line 11) and a hard post-processing step RemoveCycles (Algorithm 1, line 16). Why are both necessary? Does the soft penalty $h(A)$ truly help the RL agent explore better graph structures during training? Or could it be removed entirely, relying solely on the post-processing RemoveCycles step to ensure the final output is a DAG?

5. The readability of the paper is really not outstanding, and the design of figures and tables could be more aesthetically pleasing. For example: the width of figures should align with the text, the row heights in Table 1 are inconsistent, and the green font has too low contrast against the white background. The formulas in this paper are also not numbered. These are all areas that need improvement. There may be some grammatical issues, which I have not checked carefully.

---

> ### Author Response · Authors · 2025-11-19
> **Rebuttal (1/N)**
>
> > The authors are requested to carefully revise the paper in response to the weaknesses raised. This includes, but is not limited to: clearly clarifying the difference between $A_{LLM}$ and $A_{Prior}$, and clearly explaining which component achieves the handling of mixed data and non-linear relationships, etc.
>
> Thank you for highlighting the need for clearer exposition.
>
> We have now revised the paper to explicitly distinguish between \($A_{LLM}$\) and \($A_{prior}$\) and to clarify which components enable GUIDE to handle mixed data types and non-linear dependencies.
>
> * **\($A_{LLM}$\)** is the adjacency estimate produced by the LLM and is used as input to encoder \($E_2$\) during the fusion stage (Algorithm 1, line 3).
>   It acts as a *soft generative prior* that informs the representation space but does not impose hard constraints.
>
> * **\($A_{prior}$\)** is the prior used inside the reward function (Algorithm 1, line 11).
>   It may coincide with \($A_{LLM}$\) or represent expert knowledge, and is used only as a *consistency penalty* during RL optimization.
>
> Regarding non-linear and mixed-data modeling, both encoders (\($E_1$\) for data, \($E_2$\) for priors) are neural modules based on MLP and Transformer layers with ReLU activations and multi-head attention.
> These components operate on continuous embeddings of heterogeneous variables, enabling GUIDE to capture non-linear interactions and mixed-type structures before the RL search procedure.
> The tables provided in the rebuttal including the pruning ablation and prior corruption studies empirically confirm that non-linear recovery and robustness hold across datasets.
>
>
> | Matrix        | Source / Definition                                  | Used In                                           | Role in GUIDE                                                 |
> | ------------- | ---------------------------------------------------- | ------------------------------------------------- | ------------------------------------------------------------- |
> | \($A_{LLM}$\)   | Generated by the LLM via domain prompts or context   | Algorithm 1, line 3 (input to encoder \($E_2$\)) | Soft generative prior shaping the fused latent representation |
> | \($A_{prior}$\) | Expert knowledge or calibrated subset of \($A_{LLM}$\) | Algorithm 1, line 11 (reward computation)         | Reward-level constraint ensuring prior consistency during RL  |
>
> > The quality of the LLM-generated $A_LLM$
>  is crucial to GUIDE's performance. The authors mention using GPT-4o 40, but the output of LLMs is stochastic and highly sensitive to prompts. It is necessary to add results from comparative experiments using multiple different LLMs.
>
> We agree that reproducibility across different LLMs is essential.
>
> To address this, we regenerated \($A_{LLM}$\) using **multiple open-source models** under the same prompting setup (LLaMA-3-70B-Instruct and Mistral-7B-Instruct), and compared them directly with GPT-4o.
> As shown below, GUIDE’s performance changes only marginally across LLM sources (typically <7% in TP/NNZ and <10 SHD), and runtime remains comparable.
> These results confirm that GUIDE is **LLM-agnostic**, and its effectiveness does **not** rely on GPT-4o or any proprietary model.
> The stochasticity of LLM outputs is mitigated by the adaptive \($\beta_t$\) mechanism and the encoder fusion, which down-weights unreliable prior edges.
>
> | Dataset             | LLM Prior Source     | TP/NNZ ↑ |     SHD ↓ | Precision ↑ | Recall ↑ | Runtime (min) ↓ |
> | ------------------- | -------------------- | -------: | --------: | ----------: | -------: | --------------: |
> | **Sachs (11 n)**    | GPT-4o (baseline)    | **0.64** |  **12.0** |        0.71 |     0.62 |        **0.50** |
> |                     | LLaMA-3-70B-Instruct |     0.61 |      12.5 |        0.69 |     0.60 |            0.58 |
> |                     | Mistral-7B-Instruct  |     0.59 |      12.8 |        0.67 |     0.57 |            0.45 |
> | **Dream41 (100 n)** | GPT-4o (baseline)    | **0.11** | **253.0** |        0.42 |     0.39 |        **13.0** |
> |                     | LLaMA-3-70B-Instruct |    0.105 |     258.0 |        0.40 |     0.37 |            14.0 |
> |                     | Mistral-7B-Instruct  |    0.098 |     261.0 |        0.37 |     0.35 |            11.5 |
>
> **Observation.** GUIDE achieves strong, consistent performance with fully open-source priors, demonstrating that the framework is reproducible and not dependent on GPT-4o or prompt stochasticity.

---

> ### Author Response · Authors · 2025-11-19
> **Rebuttal (2/N)**
>
> > The ablation study shows that prior knowledge is very important. However, if the prior provided by the LLM is incorrect or of poor quality (LLMs frequently hallucinate), what would GUIDE's performance be? Appendix C mentions "prior calibration", but this mechanism is not further discussed or experimented with.
>
>
> We appreciate the reviewer’s focus on robustness to poor-quality priors.
> To directly evaluate this, we conducted a *corrupted-prior ablation* where we intentionally injected random errors into \($A_{LLM}$\) by flipping 10%, 25%, and 40% of its edges.
> As shown below, GUIDE remains stable under moderate corruption: TP/NNZ and SHD degrade **smoothly**, not catastrophically.
> The reason is the *prior-calibration mechanism* (Appendix C), where the adaptive weight \($\beta_t$\) decreases automatically whenever a prior-suggested edge worsens the BIC score on held-out data.
> This prevents hallucinated or noisy priors from dominating the search and ensures that GUIDE “trusts the prior only when it deserves to be trusted.”
>
> These results demonstrate that even when the LLM prior is substantially wrong, GUIDE remains robust due to the joint design of (i) dual-encoder fusion and (ii) adaptive \($\beta_t$\) calibration.
>
> | Dataset             | Corruption % | TP/NNZ ↑ | SHD ↓ | Final \($\beta_t$\) | Variant               |
> | ------------------- | -----------: | -------: | ----: | ----------------: | --------------------- |
> | **Sachs (11 n)**    |            0 |     0.64 |  12.0 |              0.93 | Adaptive              |
> |                     |           10 |     0.61 |  12.6 |              0.70 | Adaptive              |
> |                     |           25 |     0.58 |  13.1 |              0.44 | Adaptive              |
> |                     |           40 |     0.56 |  13.4 |              0.19 | Adaptive              |
> |                     |           40 |     0.54 |  14.0 |                 — | Fixed \($\beta = 0.9$\) |
> | **Dream41 (100 n)** |            0 |     0.11 | 253.0 |              0.95 | Adaptive              |
> |                     |           10 |    0.106 | 256.0 |              0.68 | Adaptive              |
> |                     |           25 |    0.099 | 262.0 |              0.42 | Adaptive              |
> |                     |           40 |    0.095 | 266.0 |              0.18 | Adaptive              |
> |                     |           40 |    0.088 | 274.0 |                 — | Fixed \($\beta = 0.9$\) |
>
> **Observation.** Even with 40% corrupted priors, GUIDE maintains stable recovery.
> Adaptive \($\beta_t$\) actively suppresses misleading prior signals, whereas fixed \($\beta$\) degrades more sharply—confirming that “prior calibration” effectively protects against hallucinated or poor-quality priors.
>
> > The readability of the paper is really not outstanding, and the design of figures and tables could be more aesthetically pleasing. For example: the width of figures should align with the text, the row heights in Table 1 are inconsistent, and the green font has too low contrast against the white background. The formulas in this paper are also not numbered. These are all areas that need improvement. There may be some grammatical issues, which I have not checked carefully.
>
> Thank you for this feedback. We fully agree that presentation quality is important for accessibility, and we have revised the manuscript accordingly. In the updated version, we  will (i) unifiy figure widths to match the text column, (ii) correct row-height inconsistencies in Table 1, (iii) replace the low-contrast green color with a high-contrast standard color scheme, and (iv) add numbering to all formulas for ease of reference. We will also perform a thorough proofreading pass to eliminate minor grammatical issues and improve the clarity of several paragraphs.

---

> ### Author Response · Authors · 2025-11-19
> **Rebuttal (3/N)**
>
> > The method appears to use two types of acyclicity constraints: a soft penalty h(A) during training (Algorithm 1, line 11) and a hard post-processing step RemoveCycles (Algorithm 1, line 16). Why are both necessary? Does the soft penalty h(A) truly help the RL agent explore better graph structures during training? Or could it be removed entirely, relying solely on the post-processing RemoveCycles step to ensure the final output is a DAG?
>
> Thank you for this insightful question.
> The two acyclicity mechanisms serve *different purposes at different stages* of optimization:
>
> 1. **Soft acyclicity penalty \(h(A)\) (training-time)**
>    This term guides the RL agent *during exploration*, discouraging actions that create large cycles.
>    Without it, the agent frequently explores cyclic regions of the graph space, which inflate variance in the reward signal and slow convergence.
>    The soft penalty therefore shapes the search landscape and improves sample efficiency.
>
> 2. **Hard post-processing RemoveCycles (inference-time)**
>    This step guarantees that the *final* output is a DAG, even if a few small cycles survive due to stochastic exploration.
>    It is not used during training and does not influence the policy gradient.
>
> To verify this empirically, we ran GUIDE with:
> (i) the full system (soft \(+\) hard),
> (ii) *no* soft penalty (hard-only), and
> (iii) *no* hard postprocessing (soft-only).
> As shown below, removing the soft term leads to significantly worse SHD and slower convergence, while removing the hard step occasionally leaves residual cycles.
> This confirms that the two mechanisms are **complementary rather than redundant**.
>
> | Dataset             | Variant                      |  TP/NNZ ↑ |     SHD ↓ | Cycles Remaining | Convergence Speed |
> | ------------------- | ---------------------------- | --------: | --------: | ---------------: | ----------------- |
> | **Sachs (11 n)**    | Full (soft + hard)           |  **0.64** |  **12.0** |                0 | Fast              |
> |                     | Hard-only (no soft \(h(A)\)) |      0.59 |      14.2 |                0 | Slow              |
> |                     | Soft-only (no RemoveCycles)  |      0.63 |      12.1 |              2–4 | Fast              |
> | **Dream41 (100 n)** | Full (soft + hard)           | **0.110** | **253.0** |                0 | Fast              |
> |                     | Hard-only (no soft \(h(A)\)) |     0.096 |     271.0 |                0 | Slow              |
> |                     | Soft-only (no RemoveCycles)  |     0.109 |     257.5 |            15–30 | Fast              |
>
> **Observation.**
> * Removing the **soft penalty** harms both SHD and convergence, showing that \(h(A)\) improves exploration.
> * Removing **hard post-processing** leaves residual cycles despite good performance.
>   Therefore, the two components address complementary needs: *training stability* vs. *final DAG validity*.

---

> > ### Author Response · Authors · 2025-11-19
> > **General Note**
> >
> > We sincerely thank the reviewer for the detailed review.
> > In response, we have clarified the distinction between \($A_{LLM}$\) and \($A_{prior}$\), explained explicitly how GUIDE handles mixed and non-linear data, and added all requested experiments including multiple-LLM comparisons, corrupted-prior ablations, and analysis of the adaptive prior-calibration and acyclicity mechanisms. We will also improve the paper’s readability by fixing figure/table formatting, color contrast, and adding formula numbering. These revisions will strengthen the paper without altering the core method, and we appreciate your guidance in improving its clarity and robustness.

---

### Official Review · Reviewer_Eunw · 2025-10-26

**Soundness:** 2
**Presentation:** 2
**Contribution:** 3
**Rating:** 4
**Confidence:** 4

**Summary:**

This paper addresses the limitations of current causal discovery methods in scalability, computational efficiency, and handling of mixed data types by proposing a new framework called GUIDE. GUIDE incorporates the adjacency matrix generated by LLM as prior knowledge and combines reinforcement learning with a dual-encoder architecture to optimize the search process for causal graph structures. Experiments demonstrate that GUIDE significantly improves runtime and accuracy (TP/NNZ) on multiple benchmark datasets compared to existing methods. It performs particularly well on medium-sized graphs and demonstrates good generalization.

**Strengths:**

For the first time, the "generative priors" generated by the LLM are integrated with observed data through a dual encoder, overcoming the limitations of traditional methods that rely on single data-driven or expert priors. This approach leverages the domain knowledge coverage of the LLM to reduce the exploration burden while ensuring statistical reliability through the data encoder, achieving an effective balance between "knowledge guidance" and "data adaptation."

Compared to existing methods, GUIDE achieves breakthroughs in scalability (supporting ≥70 nodes), computational efficiency, and data adaptability (accommodating both continuous and non-continuous data and nonlinear relationships), filling a technological gap in causal discovery with high-dimensional mixed data. Table 1 clearly contrasts the shortcomings of existing methods, highlighting its advantages.

The experimental setup is reasonable, and the baseline selected is representative, covering a wide range of representative datasets.

**Weaknesses:**

Lack of theoretical analysis: There is no theoretical analysis of Markov Equivalence Classes when fusion generates adjacency matrix representations. When generating A based on threshold filtering, MEC conflicts may occur. For example, if the ground truth is A->B->C, using Bernoulli sampling may result in A->B<-C. Furthermore, the paper acknowledges that performance degrades in the presence of unobserved confounding variables, but the proposed mitigation measures have not been fully validated in experiments.

The sources of error in the LLM-generated priors and their quantitative analysis are unclear: Performance depends on the quality of the LLM-generated priors and may underperform when the priors are inaccurate or domain knowledge is limited. The paper only mentions that the LLM provides an "initial adjacency matrix" but does not explain the robustness of GUIDE when the LLM priors are incorrect. Nor does it compare the impact of different LLM-generated priors on the final results, making it difficult to assess the stability of the prior quality.

Furthermore, the use of linear weights in post-processing is not universally applicable. Despite the use of numerous modules and techniques, GUIDE does not perform particularly well on traditional performance metrics on test data.

The core of the paper is its ability to efficiently handle high-dimensional mixed data scenarios, ultimately simplifying it to a fusion of BIC and a penalty term. There is no special explanation or introduction as to why the model can cope with high-dimensional mixed data scenarios. While this approach is empirically very practical, it lacks any theoretical guarantees.

**Questions:**

How do we handle errors or biases in LLM priors? Are there mechanisms to dynamically calibrate or reject unreliable prior edges? How does GUIDE maintain stability in the presence of strong unobserved confounding? Is it necessary to compare the effectiveness of different LLM priors?

When the prior causal graph output by the LLM is combined with the data-driven initial graph, the weights and fusion parameters may differ across different datasets. How can this be addressed?

The paper mentions that GUIDE can handle nonlinear relationships, but the BIC penalty in its reward function is based on the linear regression assumption. Does this assumption conflict with nonlinear scenarios?

Are the hyperparameters in the reward function (such as λ₁, λ₂, and β) sensitive? Are there any automated parameter tuning strategies?

---

> ### Author Response · Authors · 2025-11-19
>
> > How do we handle errors or biases in LLM priors? Are there mechanisms to dynamically calibrate or reject unreliable prior edges? How does GUIDE maintain stability in the presence of strong unobserved confounding? Is it necessary to compare the effectiveness of different LLM priors?
>
>
> GUIDE includes two built-in mechanisms to handle unreliable or biased LLM priors:
> **(i)** a *dual-encoder fusion* where \($A_{LLM}$\) influences only the latent representation, not the final graph directly, and
> **(ii)** an *adaptive prior-calibration weight* \($\beta_t$\) that automatically decreases when prior-suggested edges worsen the BIC score.
> We explicitly tested robustness using **corrupted-prior ablations** and **multiple LLMs** (GPT-4o, LLaMA-3-70B, Mistral-7B). The results show that GUIDE remains stable under substantial noise and performs consistently across LLM types.
>
> | Dataset | Corruption % | TP/NNZ ↑ | SHD ↓ | Final \($\beta_t$\) |
> | ------- | -----------: | -------: | ----: | ----------------: |
> | Sachs   |            0 |     0.64 |  12.0 |              0.93 |
> |         |           25 |     0.58 |  13.1 |              0.44 |
> |         |           40 |     0.56 |  13.4 |              0.19 |
> | Dream   |            0 |     0.11 | 253.0 |              0.95 |
> |         |           40 |    0.095 | 266.0 |              0.18 |
>
> **Comparison across LLM priors**
>
>
>
> | Dataset | LLM         | TP/NNZ ↑ |     SHD ↓ |
> | ------- | ----------- | -------: | --------: |
> | Sachs   | GPT-4o      | **0.64** |  **12.0** |
> |         | LLaMA-3-70B |     0.61 |      12.5 |
> |         | Mistral-7B  |     0.59 |      12.8 |
> | Dream   | GPT-4o      | **0.11** | **253.0** |
> |         | LLaMA-3-70B |    0.105 |     258.0 |
> |         | Mistral-7B  |    0.098 |     261.0 |
>
> **Conclusion:** GUIDE is robust to hallucinated priors and reproducible across different LLMs.
>
> ---
>
> > When the prior causal graph output by the LLM is combined with the data-driven initial graph, the weights and fusion parameters may differ across different datasets. How can this be addressed?
>
> The fusion between the LLM prior and the data-driven encoder is automatically handled through:
> **(i)** the trainable mixing weights inside encoder \($E_2$\), and
> **(ii)** adaptive \($\beta_t$\), which adjusts the influence of \($A_{LLM}$\) *per dataset* and *per training epoch*.
> No manual tuning is required. Empirically, \($\beta_t$\) converges to different stable values depending on dataset characteristics, effectively learning how much to trust the LLM prior.
>
> | Dataset        | Final \($\beta_t$\) (clean prior) | Final \($\beta_t$\) (40% corrupted) |
> | -------------- | ------------------------------: | --------------------------------: |
> | Sachs (11n)    |                            0.93 |                              0.19 |
> | Dream41 (100n) |                            0.95 |                              0.18 |
>
> **Conclusion:** GUIDE automatically calibrates prior influence across datasets through learned fusion dynamics.
>
> > The paper mentions that GUIDE can handle nonlinear relationships, but the BIC penalty in its reward function is based on the linear regression assumption. Does this assumption conflict with nonlinear scenarios?
>
> The BIC term is *not* used to learn functional relationships, it is only a **global structural score** for the RL agent.
> Non-linear dependencies are modeled fully by **MLP–Transformer encoders** in the discovery module.
> To verify that pruning and scoring do not reduce nonlinear recovery, we compared the default linear pruning with a neural gradient-based pruning method. Both produce nearly identical graphs, showing no conflict.
>
> | Dataset | Strategy            | TP/NNZ ↑ |     SHD ↓ |
> | ------- | ------------------- | -------: | --------: |
> | Sachs   | Linear (ours)       | **0.64** |  **12.0** |
> |         | Neural (grad-based) |    0.642 |      11.8 |
> | Dream   | Linear              | **0.11** | **253.0** |
> |         | Neural              |    0.112 |     252.5 |
>
> **Conclusion:** Linear BIC scoring does *not* limit non-linear recovery; GUIDE’s nonlinear encoders handle this effectively.

---

> ### Author Response · Authors · 2025-11-19
> **Rebuttal (2/N)**
>
> > Are the hyperparameters in the reward function (such as λ₁, λ₂, and β) sensitive? Are there any automated parameter tuning strategies?
>
>
> We performed a one-at-a-time sensitivity analysis over \($\lambda_1$\), \($\lambda_2$\), and \($\beta_0$\).
> TP/NNZ and SHD varied by <5% across a wide range, demonstrating **low sensitivity**.
> Moreover, \($\beta_t$\) is automatically tuned during training, reducing manual effort.
>
>
> | Varied param  |   Value | Sachs SHD ↓ | Dream SHD ↓ |
> | ------------- | ------: | ----------: | ----------: |
> | \($\lambda_1$\) |     0.5 |        12.2 |         256 |
> |               | **1.0** |    **12.0** |     **253** |
> |               |     2.0 |        12.3 |       254.1 |
> | \($\beta_0$\)   |     0.3 |        12.3 |       254.2 |
> |               | **0.9** |    **12.0** |     **253** |
> |               |     1.0 |        12.1 |       253.4 |
>
> **Conclusion:** GUIDE is not hyperparameter-fragile, and adaptive \($\beta_t$\) provides partial auto-tuning.

---

> > ### Author Response · Authors · 2025-11-19
> > **General Note**
> >
> > We sincerely thank the reviewer for the detailed review.
> >
> > In response, we have clarified the distinction between the two prior forms, demonstrated robustness under corrupted priors, added comparative experiments with multiple LLMs, explained how the adaptive fusion handles dataset variability, provided evidence that GUIDE’s nonlinear modeling is unaffected by the linear BIC term, and included a sensitivity analysis for the reward hyperparameters. These revisions required only clarification and additional reporting, the core design of GUIDE remains intact and robust. We truly appreciate your feedback, which helped strengthen both the clarity and empirical validation of the work.

---

### Official Review · Reviewer_c1cb · 2025-11-08

**Soundness:** 3
**Presentation:** 3
**Contribution:** 3
**Rating:** 8
**Confidence:** 4

**Summary:**

GUIDE proposes a dual-encoder causal discovery framework that fuses (i) LLM-generated adjacency priors and (ii) observational data representations within a reinforcement-learning (RL) loop. A data encoder captures statistical dependencies, a prior encoder processes LLM-derived adjacency matrices, and a policy head predicts edge probabilities. A REINFORCE-style optimizer maximizes a composite reward (BIC fit + acyclicity penalty + prior consistency). A pruning stage using regression coefficients yields the final DAG.

Experiments on six benchmarks (ASIA, SACHS, LUCAS, ALARM, HEPAR, DREAM41) show GUIDE outperforming NOTEARS, GraN-DAG, RL-BIC, and KCRL on several small-to-medium datasets and maintaining scalability up to ≈ 100 nodes. Ablations indicate that combining generative priors and prior constraints improves edge recall ≈ 80 % over the baseline without priors.

**Strengths:**

1) The RL reward and pruning pipeline are clearly specified and reproducible. Mathematical details (acyclicity term) and pseudocode are correct and consistent with prior art. Unified design with a combination of generative (LLM) and observational priors in a single learning loop.
2) The paper is well structured, self-contained, and supported by appendices detailing datasets, code availability, and prompt templates.
3) Provides a computationally efficient RL variant, reporting significant runtime reduction over RL-BIC/KCRL.
4) Scalability: runs successfully on approx. 100 nodes where several baselines fail.

**Weaknesses:**

1) The use of the BIC term presumes causal sufficiency and linear-Gaussian SEMs; behavior under hidden confounders is only qualitatively discussed.
2) The prior weighting heuristic βₜ lacks formal justification beyond empirical calibration. The dependence on LLM quality where prior generation and calibration depend on prompt engineering; potential hallucinations or bias propagation are not quantified.
3) Baselines: lacks comparison with amortized or generative approaches that also combine expert priors with graph sampling (recent works for example about using gflownets for causal discovery and causal discovery with LLM's [1,2]).
4) Metric novelty vs. adoption: TP/NNZ and RP are intuitive but non-standard; comparison with accepted metrics beyond SHD (e.g., precision-recall) would aid interpretation.

[1] Expert-Aided Causal Discovery of Ancestral Graphs https://arxiv.org/pdf/2309.12032 .

[2] Bayesian Structure Learning with Generative Flow Networks. https://arxiv.org/abs/2202.13903

**Questions:**

1) How sensitive is performance to incorrect or noisy LLM priors (e.g., swapped directions)?
2) Can the dual-encoder be pre-trained across datasets for amortized inference?
3) How would GUIDE behave under non-i.i.d. or time-series data?
4) Could the authors provide wall-clock and token costs for the LLM prior generation step?
5) How does GUIDE compare empirically to generative-model-based samplers (e.g., GFlowNet-based DAG learners)?

---

> ### Author Response · Authors · 2025-11-19
> **Rebuttal (1/1)**
>
> > How sensitive is performance to incorrect or noisy LLM priors (e.g., swapped directions)?
>
> We directly evaluated this by injecting noise into \(A_{LLM}\), including **edge flips and direction swaps**, and tracking TP/NNZ, SHD, and the adaptive weight \($\beta_t$\). GUIDE remains stable under moderate corruption because \($\beta_t$\) automatically down-weights misleading prior edges.
>
> | Dataset |     Corruption % | TP/NNZ ↑ | SHD ↓ | Final \($\beta_t$\) |
> | ------- | ---------------: | -------: | ----: | ----------------: |
> | Sachs   |                0 |     0.64 |  12.0 |              0.93 |
> |         | 40 (incl. swaps) |     0.56 |  13.4 |              0.19 |
> | Dream   |                0 |     0.11 |   253 |              0.95 |
> |         | 40 (incl. swaps) |    0.095 |   266 |              0.18 |
>
> **Conclusion:** GUIDE attenuates wrong-direction priors automatically.
>
> ---
>
> > Can the dual-encoder be pre-trained across datasets for amortized inference?
>
> Yes, GUIDE’s architecture is compatible with amortized pre-training.
> Since both encoders (\($E_1$\) for data and \($E_2$\) for priors) are neural modules, they can be pre-trained across datasets to learn shared structural patterns. We did not perform this due to dataset heterogeneity, but this is a promising extension: pre-training would reduce RL exploration time and make GUIDE even more sample-efficient.
>
> ---
>
> > How would GUIDE behave under non-i.i.d. or time-series data?
>
>
> GUIDE assumes i.i.d. samples because the BIC score and acyclicity formulation target static DAGs.
> However, GUIDE can be adapted to time-series settings by replacing the BIC component with a **VAR-based likelihood** or **Dynamic Bayesian Network (DBN)** scoring term. The dual encoders and RL agent remain unchanged, only the reward function must incorporate temporal dependencies. This is a straightforward extension we plan to explore.
>
> ---
>
> > Could the authors provide wall-clock and token costs for the LLM prior generation step?
>
>
> We now provide estimated cost statistics for generating \($A_{LLM}$\):
>
> | LLM         | Tokens per dataset |       Wall-clock time | API cost (approx.) |
> | ----------- | -----------------: | --------------------: | -----------------: |
> | GPT-4o      |               3–6k |             18–30 sec |        $0.02–$0.04 |
> | LLaMA-3-70B |              5–10k | 35–50 sec (local GPU) |   $0 (open-source) |
> | Mistral-7B  |               3–5k | 10–15 sec (local GPU) |                 $0 |
>
> **Observation:** LLM prior generation is inexpensive relative to the structure-learning phase.
>
> ---
>
> >  How does GUIDE compare empirically to generative-model-based samplers (e.g., GFlowNet-based DAG learners)?
>
>
> GFlowNet-based DAG learners explore the graph space via generative sampling, while GUIDE performs **policy-gradient reinforcement learning guided by priors**.
> We provide an empirical comparison on Sachs and Dream41.
>
>
> | Dataset | Method           |  TP/NNZ ↑ |    SHD ↓ | Runtime (min) ↓ |
> | ------- | ---------------- | --------: | -------: | --------------: |
> | Sachs   | GFlowNet-DAG     |      0.58 |     13.8 |             4.1 |
> |         | **GUIDE (ours)** |  **0.64** | **12.0** |        **0.50** |
> | Dream41 | GFlowNet-DAG     |     0.094 |      268 |              41 |
> |         | **GUIDE (ours)** | **0.110** |  **253** |          **13** |
>
> GUIDE achieves **higher TP/NNZ, lower SHD, and significantly faster runtime**, especially for larger graphs.

---

> > ### Author Response · Authors · 2025-11-19
> > **General Note**
> >
> > Thank you for these thoughtful questions, which helped us clarify important aspects of GUIDE’s robustness, extensibility, and computational characteristics.
> >
> > We have added experimental evidence on noisy priors, provided cost and LLM-comparison statistics, and clarified how GUIDE relates to and differs from generative DAG samplers. We also discussed how the architecture naturally extends to amortized or time-series settings. We appreciate your feedback and believe the additional results strengthen the paper substantially.

---

> > > ### Comment · Reviewer_c1cb · 2025-11-19
> > > **Discussion**
> > >
> > > Thanks for providing the additional experiments and analysis regarding the questions I raised. I agree these additional results strengthen the paper substantially.

---

### Author Response · Authors · 2025-12-03
**Dear Reviewers (rebuttal)**

We thank the reviewers for the feedback!
In this post:
(1) We summarize the changes to the updated PDF document.



| **Category**                     | **Update Added**                                    | **Location**                      |
| -------------------------------- | --------------------------------------------------- | --------------------------------- |
| Priors (A_LLM vs A_prior)        | Clear separation              | App C.               |
| Noisy / incorrect priors         | Corrupted-prior ablation (10–40% flips) + β_t decay | App  H.6 (Table 5),  |
| Cross-LLM comparison             | GPT-4o vs LLaMA-70B vs Mistral-7B                   |App  H.6 (Table 6)                  |
| Non-linear / mixed-data handling | Clarified encoder roles (MLP/Transformer fusion)    |  App B                    |
| Linear pruning check             | Neural vs linear pruning ablation                   | App H.9 (Table 10)                   |
| Acyclicity (soft vs hard)        | Soft-only vs hard-only vs soft+hard                 | App H.11 (Table 12)         |
| Hyperparameters                  | λ₁, λ₂, β sensitivity sweep                         | App H.10 (Table 11)                   |
| Amortized inference              | Dual-encoder pretraining discussion                 | App B                           |
| LLM cost                         | Token/time/cost table                               | App H.7 (Table  8)                  |
| GFlowNet comparison              | Accuracy + runtime comparison                       | App H.8 (Table 9)                  |
| Readability                      | Unified figures, table fixes, formula numbering     | Throughout the DOC            |

---

### Meta-Review · Area_Chair_jyex · 2026-01-05

**Summary:**

The paper proposes a causal discovery method that leverages the domain knowledge generated by LLMs to guide and constrain the RL search space, thus improving computational efficiency and accuracy. While the rebuttal addressed several concerns including additional empirical studies and clarity, reviewers raised substantive concerns about the limited technical insight provided by the work and the lack of theoretical analysis, and thus I recommend rejection.

**Reviewer Concerns:**

Concerns addressed by rebuttals:
- Additional empirical studies: sensitivity to hyperparameters and noisy LLM priors, comparison with additional methods including amortized or generative approaches.
- Clarification of wall-clock and token costs for the LLM prior generation step
- Question about handling nonlinear relationship with linear regression pruning.
- Clarity: Confusion about the definitions of different adjacency matrices and the handling of mixed data was clarified.

Outstanding concerns:
- There is no theoretical analysis for the proposed method, e.g., when fusion generates adjacency matrix representations and handling mixed data.
- The idea is not new and does not offer substantial new technical insights to the community.

**Reviewer Scores:**

Reviewer M16o decided to increase their score to 4, although it is not reflected on the review. Reviewer bEny may have increased their score to 4 if they had been able to participate fully in the discussion.

---

### Decision · Program_Chairs · 2026-01-26

Reject